# LARP3, LARP7, and MePCE are involved in the early stage of human telomerase RNA biogenesis

Tsai-Ling Kao [ORCID][1], Yu-Cheng Huang [ORCID][1], Yi-Hsuan Chen [ORCID][1], Peter Baumann [ORCID][2,3] & Chi-Kang Tseng [ORCID][1] ✉

Human telomerase assembly is a highly dynamic process. Using biochemical approaches, we find that LARP3 and LARP7/MePCE are involved in the early stage of human telomerase RNA (hTR) and that their binding to RNA is destabilized when the mature form is produced. LARP3 plays a negative role in preventing the processing of the 3′-extended long (exL) form and the binding of LARP7 and MePCE. Interestingly, the tertiary structure of the exL form prevents LARP3 binding and facilitates hTR biogenesis. Furthermore, low levels of LARP3 promote hTR maturation, increase telomerase activity, and elongate telomeres. LARP7 and MePCE depletion inhibits the conversion of the 3′-extended short (exS) form into mature hTR and the cytoplasmic accumulation of hTR, resulting in telomere shortening. Taken together our data suggest that LARP3 and LARP7/MePCE mediate the processing of hTR precursors and regulate the production of functional telomerase.

Telomerase is a ribonucleoprotein complex that contains two highly conserved components in its catalytic core. In humans, one core component is a noncoding RNA called human telomerase RNA (hTR); the other is a protein enzyme called human telomerase reverse transcriptase (hTERT). hTERT copies the template region within hTR to replenish telomeric DNA sequences. Thus, the ends of chromosomes are protected, and the lengths of telomeres are maintained[1]. hTR is transcribed by RNA polymerase II and accumulates in cells as a 451-nt-long RNA[2]. Longer forms of this transcript have been reported[3,4]. Accumulating evidence suggests that longer transcripts are predominantly degraded by RNA exosomes via a process mediated by the cap-binding complex (CBC) and the nuclear exosome targeting (NEXT) complex[5]. However, a fraction of the long transcripts can be processed into the mature form[5]. During hTR maturation, 3′-end processing is mediated in concert with multiple 3′-to-5′ exonucleases[3,6,7]. The 3′-extended long form (exL, ≥ 460-nt hTR) of hTR is first trimmed by the exosomal component RRP6 to produce the 3′-extended short form (exS, ≥ 452-nt and ≤460-nt hTR)[3]. The exS of hTR is then processed by two other exonucleases (PARN and TOE1) that function in parallel and/ or sequentially to produce mature 451-nt hTR[5–8]. The requirement of RRP6 and PARN for hTR 3′ end maturation has been supported by an in vitro cell-free system[3]. Knockdown of RRP6 and PARN impaired the 3′ end processing of hTR[3]. Given that PARN is detected mainly in the nucleolus and that TOE1 is located in Cajal bodies[7], the formation of mature hTR has been suggested to couple to 3′-end processing with RNA trafficking.

Human telomerase assembly proceeds via precise stepwise binding of protein components to hTR during 3′-end maturation[3,5,9]. The structure of hTR is highly organized and plays a role in mediating its stability, trafficking, and maturation[10]. The 3′-domain of hTR folds into a box H/ACA-like domain[11], which is bound by the box H/ACA complex[12]. The assembly of the pre-H/ACA complex on hTR via cotranscription is thought to be critical for protecting longer transcripts from rapid degradation[5,9]. A biochemical study showed that the exL form of hTR folds into a triple-helix structure[3]. Although how the triple helix conformation transiently protects the exL form of hTR from rapid degradation remains unclear, this structure provides an opportunity for the H/ACA complex to bind[3]. The binding of the H/ACA complex is not only critical for hTR stability but also attenuates processing by PARN at position 451[3]. Mutations in most well-characterized components of

[1]Department of Microbiology, College of Medicine, National Taiwan University, Taipei, Taiwan. [2]Institute of Developmental Biology and Neurobiology, Johannes Gutenberg University, Mainz, Germany. [3]Institute of Molecular Biology, Mainz, Germany. ✉e-mail: ckt0513@ntu.edu.tw

human telomerase and telomeres, as well as their accessory factors, have been reported in premature-ageing diseases, such as dyskeratosis congenita, Hoyeraal–Hreidarsson syndrome, aplastic anaemia, and idiopathic pulmonary fibrosis[13–16].

At the molecular level, patients with these telomere biology disorders (TBDs) exhibit accelerated telomere shortening. In addition to mutations in canonical TBDs, mutations in several other genes have been associated with dysregulation of telomere maintenance in human diseases[17]. One of these dysregulated gene products is in the La-related protein (LARP) family, which is an important RNA-binding protein family that emerged early in eukaryote evolution and is involved in a wide range of crucial functions in cells involving both coding and noncoding RNAs[18,19]. Seven distinct LARP-encoding genes have been identified in humans. Among human LARPs, LARP3 (a genuine La protein) and LARP7 have previously been implicated in human telomere maintenance[20,21]. Aberrantly expression of LARP3 has been found in various cancer types, including chronic myelogenous leukaemia (CML)[22]. LARP3 has been shown to interact directly with hTR and cause telomere shortening when exogenous LARP3 is overexpressed in certain cell lines[20]. Whether LARP3 is involved in hTR biogenesis remains unclear.

A total of 52 pathogenic variants in the *LARP7* gene have been identified in patients with Alazami syndrome[23]. Patients with Alazami syndrome exhibit very short telomeres, and LARP7 knockdown in cancer cells causes a reduction in telomerase activity and telomere shortening[21]. In ciliated protozoa and fission yeast, the LARP family protein p65 in *Tetrahymena thermophila*[24], p43 in *Euplotes aediculatus*[25], and Pof8 in *Schizosaccharomyces pombe*[26–28] are constitutive components of active telomerase. In addition to LARP7, MePCE, a LARP7-interacting protein, has been implicated in neurodevelopment[29]. The heterozygous methyl phosphate capping enzyme (MePCE) nonsense variant c.1552 C > T/p. (Arg518*) has been identified[29]. Patients with MePCE mutations exhibit a neurodevelopmental disorder phenotype similar to that of patients with loss-of-function mutations in *LARP7*. Bin3/MePCE 1 (Bmc1) is the putative fission yeast orthologue of MePCE and cooperates with Pof8 to recognize correctly folded TER1[30,31]. Whether human LARP7 and MePCE are involved in hTR and biological functions in a manner analogous to that in other species remains unclear.

In this study, we established in vitro systems that allowed us to monitor hTR 3′-end maturation and protein component assembly. Telomerase assembled in vitro was functional. Using these systems, we found that LARP3, LARP7, and MePCE participate in the early stage of human telomerase biogenesis. LARP3 binds to the exL form of hTR before the H/ACA complex binding and prevents 3′-end processing. The triple-helix structure of the exL form of hTR prevents binding by LARP3 and facilitates 3′-end processing. Consistent with these observations, LARP3 knockdown facilitated the maturation of hTR and caused an increase in telomerase activity. Given that the expression levels of LARP3 increase during CML progression[32] and that CML patients normally exhibit short telomeres[33], our data showed that reducing LARP3 expression caused an increase in telomerase activity and telomere elongation in K562 cells. LARP7 and MePCE binding then increases as LARP3 decreases during the conversion of the exS form to the mature form. LARP7 and MePCE knockdown caused telomere shortening by affecting both hTR maturation and localization. Our data suggest that human telomerase assembly is a highly dynamic process that involves compositional and conformational rearrangement, which leads to the production of a functional telomerase.

## Results

### The establishment of in vitro systems to examine the biogenesis of human telomerase

Research on the fates of human hTR precursors is challenging due to the extremely low abundance of endogenous pre-telomerase

complexes in a cell. To overcome these limitations and to dissect the molecular mechanisms involved in hTR maturation, in vitro systems of human telomerase biogenesis are established. The in vitro systems we established to study hTR can be classified into 3 major parts: the examination of 3′-end maturation of hTR, telomerase component assembly, and telomerase activity (Supplementary Fig. 1a). To analyse the 3′-end processing of hTR, we synthesized the H/ACA domain of hTR (starting with nucleotide 206) of the exL form in vitro with oligo A tails in the presence of α-$^{32}$P-UTP. The oligoadenylated exL form of hTR was incubated with whole-293T cell extract. During the incubation period, deadenylation neared completion within 10 min (Fig. 1a, lane 2). The exS and mature forms of hTR were produced after 30–60 min and 2 h of incubation, respectively (Fig. 1a, lanes 4 and 5). Previous studies have suggested that long transcripts are predominantly degraded in vivo[5]. Consistent with these observations[5], more than 80% of the exL form was degraded, and only 20% of the exL form was converted to the mature form of hTR in the in vitro assay (Fig. 1b and Supplementary Fig. 1b). These data indicate that the 3′-end processing of hTR was successful in vitro.

For analysis of the assembly of telomerase components on hTR, telomerase complexes assembled on the biotinylated exL form of hTR (nucleotides from 1 to 461 with an oligo A tail) with a monoguanosine cap (MMG) were purified at different time points by pulling ribonucleoproteins (RNPs) down with streptavidin beads (Supplementary Fig. 1a). The purified telomerase complexes were subjected to Northern blotting using a probe matching either the 5′ or 3′ region of hTR (Supplementary Fig. 1c). Northern blotting analysis revealed that the percentage of full-length exL with a 5′ monoguanosine cap decreased by more than 80%, suggesting that the turnover rates of the full-length and short forms of hTR were similar (Supplementary Fig. 1c). When the probe matching the mature 3′ region of hTR was used, the truncated version of hTR was detected, indicating that a small fraction of exL was degraded from the 5′ end (Supplementary Fig. 1c). For analysis of the assembly of telomerase components, Western blotting was carried out (Fig. 1c). All H/ACA complex components (DKC1, NHP2, NOP10, NAF1, and GAR1) were detected. This finding supports in vivo observations[9,34] suggesting that NAF1 in the pre-H/ACA complex (NAF1-DKC1-NOP10-NHP2) binds to hTR and is subsequently replaced with GAR1 to produce a functional H/ACA complex. NAF1 was associated mainly with the exL form of hTR in the early maturation stage (Fig. 1c, lanes 9 and 10). In contrast, GAR1 was associated with hTR in the late stage, while NAF1 was disassociated from the exL form of hTR (lanes 10–12). TCAB1 appeared to associate with hTR after NAF1 disassociated during GAR1 binding (lanes 10–12). DHX36, which has been shown to interact with the 5′-guanosine tracts of hTR[35], was pulled down. The binding of DHX36 to hTR remained constant during the process of 3′ end maturation (Fig. 1c, lanes 8–12). DHX36 binds to the 5′ G-rich region of hTR. These 3′ truncated versions also contribute to the binding of DHX36.

For determination of the catalytic potential of the in vitro assembled telomerase, telomerase was assembled with exL and its mature forms, followed by the purification of streptavidin (Fig. 1d). A direct primer extension assay was performed to measure telomerase activity (Fig. 1e). When the purified telomerase contained similar amounts of DHX36, DKC1 and hTR (Fig. 1d), the telomerase assembled with the exL and mature forms showed enzymatic activity (Fig. 1e, f and Supplementary Fig. 2b). However, the enzymatic activity of the purified telomerase with mature hTR was greater than that of the exL-containing telomerase (Fig. 1f). We further examined the processivity of the purified telomerase (Fig. 1g and Supplementary Fig. 2c). Quantification analysis revealed that telomerase with the mature hTR had greater processivity than that with the exL form (Fig. 1g and Supplementary Fig. 2c). These data suggest that telomerase assembly undergoes compositional rearrangement during 3′-end maturation

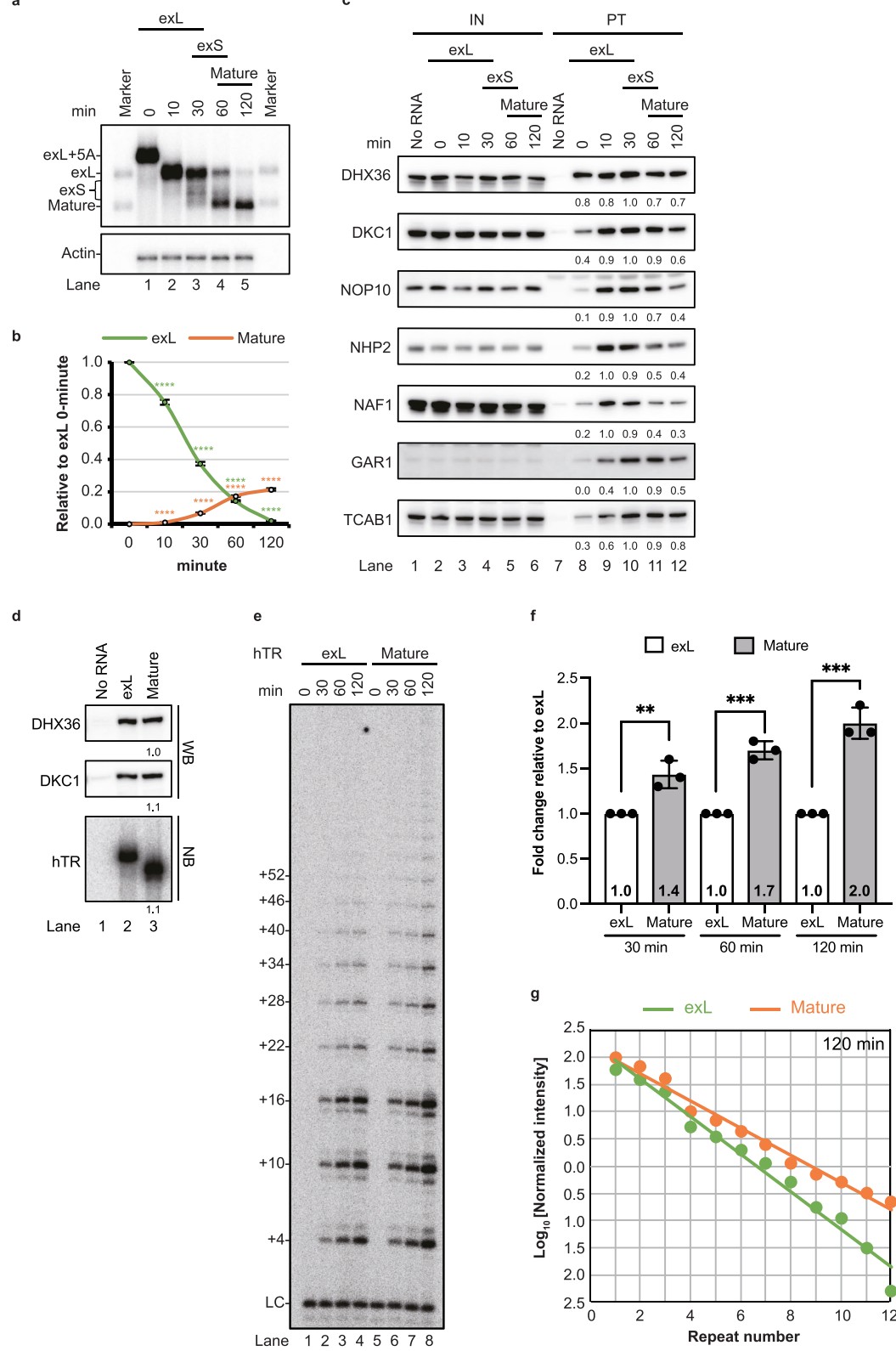

and that the removal of 3′-extended sequences from hTR may increase telomerase activity and processivity.

## LARP3, LARP7, and MePCE are involved in the early stage of telomerase assembly

Biochemical and structural studies of telomerases from ciliated protozoa and fission yeast revealed that a La-related protein and its interacting partners are the constitutive components of a telomerase holoenzyme and are critical for the assembly and activity of this telomerase[26,30,36]. Therefore, we examined the associations of human La-related proteins with hTR. LARP3, LARP7 and MePCE were significantly associated with the hTR (Fig. 2a). A time course analysis revealed that LARP3 associated with the hTR concurrently with the binding of LARP7 and MePCE (Fig. 2a, lanes 8–10). All component

**Fig. 1 | The establishment of in vitro systems to examine the biogenesis of human telomerase. a** The in vitro 3′ end processing assay was carried out in 293T cell extracts at 37 °C for the indicated times. RNA was purified and resolved on a 6% polyacrylamide gel containing 8 M urea. Actin served as the loading control. **b** The exL and mature forms of hTR signals were normalized to 0 min. The bars are presented as mean values +/− SD calculated from triplicate experiments of three technical replicates. Dots represent data points from individual experiments. The significance was calculated with a two-sided Student's t test; *$p < 0.05$, **$p < 0.01$, ***$p < 0.005$, ****$p < 0.001$. **c** Western blotting analysis of telomerase assembled on biotin-labelled hTR pulled down with streptavidin beads for the indicated times. The signals from purified telomerase (lanes 7–12) were normalized to the peak signal. Input (IN) represents 10% of the purified telomerase (PT) samples. **d** Western blotting and Northern blotting analysis of the in vitro purified telomerase

assembled on exL and mature forms of hTR. **e** Telomerase activity of the in vitro purified telomerase assembled on exL and mature forms of hTR. **f** Telomerase activity at the indicated time points of telomerase assembly on the mature form was normalized to that of telomerase assembly on the exL form. The mean values +/− SD were calculated from triplicate experiments of three biological replicates. Dots represent data points from individual experiments. The significance of the change in telomerase activity between samples was calculated with a two-sided Student's t test; *$p < 0.05$, **$p < 0.01$, ***$p < 0.005$, ****$p < 0.001$. **g** Telomerase processivity quantitation of in vitro-purified telomerase assembled on exL and mature forms of hTR. The intensity of each major band (+4, +10, +16, +22, +28, and so on) from the telomerase activity assay in **e** was quantitated by phosphorimager analysis. Source data are provided as a Source data file.

binding in the telomerase complexes was destabilized when the mature hTR forms were produced (Fig. 2a, lanes 11 and 12). For determination of how the 3′-extended sequence affects component binding, different forms of hTR species were individually generated, including the exL, exS, mature forms, 3′ stem loop deletion (3′ SL del), and pseudoknot domain of hTR (Fig. 2b). Western blotting of telomerase complexes assembled with different hTR species revealed that DHX36 bound to all the forms of hTR. Consistent with previous studies showing that the 3′ stem loop is critical for H/ACA complex assembly[37], deletion of the 3′ stem loop abolished the binding of DKC1 (Fig. 2b, lane 11). LARP7 and MePCE preferentially associated with the 3′-extended forms (exL and exS forms) of hTR (Fig. 2b, lanes 8–10). The extension of the 451-nt hTR end by 5 nucleotides (nucleotides 1–455 in the exS form) and 10 nucleotides (nucleotides 1–461 in the exL form) stabilized the binding of LARP7/MePCE and LARP3, respectively (Fig. 2b, lanes 8 and 9), suggesting that a 3′-extension sequence is required for the stable binding of LARP3, LARP7, and MePCE to hTR. To determine whether LARP3, LARP7, and MePCE-associated telomerase can exhibit telomerase activity, we immunoprecipitated LARP3, LARP7, and MePCE (Fig. 2c). Immunoprecipitation with the anti-LARP3 antibody coprecipitated LARP7 and MePCE (Fig. 2c, lane 7) and vice versa (Fig. 2c, lanes 8 and 9), but did not significantly coprecipitate DKC1. Compared to DKC1, LARP3, LARP7, and MePCE coprecipitated with only small amounts of hTR (Fig. 2d). LARP3-, LARP7-, and MePCE-associated endogenous telomerases exhibited only low levels of telomerase activity (Fig. 2e, f, and Supplementary Fig. 3). Taken together, these data suggest that LARP3, LARP7, and MePCE are involved mainly in the early stage of telomerase assembly and disassociate from telomerase when functional telomerase is produced.

## LARP3 binding competes with tertiary structure formation
LARP3 preferentially associated with the exL form of hTR (Fig. 2b). The exL form of hTR is a highly organized structure that contains two stem–loop conformations and a 3′-terminal UUU stretch. The 3′-terminal UUU stretch in the exL form has been proposed to form a triple helix structure in concert with box H and the UCU sequence between the P4.2 and P5 stems[3] (Fig. 3a). In addition, the 3′-terminal U stretch, which is common at the 3′-end of RNA polymerase III transcripts, is the binding target of LARP3[38]. This finding prompted us to speculate that the preferential binding of LARP3 to the exL form is due to either the 3′-terminal UUU stretch or the structure of the triple helix (Fig. 3a). To investigate the effect of the 3′-terminal UUU sequence on LARP3 binding, we generated the U460C mutant hTR. The U460C mutant lacked the 3′-terminal UUU stretch but showed a strengthened triple-base interaction[3]. In contrast, we disrupted the tertiary base interactions but retained the 3′-terminal UUU sequence by changing the box H sequence (372-AGAGGA-377). 375-GG-376 was replaced with AU (372-AGAAUA-377), which converts the hTR H box into the equivalent motif of snoRNA U92 and partially retains the base pair between the terminal sequence and box H[3]. For complete disruption of the interaction between the 3′-terminal UUU and box H, a disease-

related box H mutant (375-377GGAdel) was generated, which was identified in patients with idiopathic pulmonary fibrosis[39] (Fig. 3a).

The results showed that the U460C hTR mutant pulled down less LARP3 than the wild-type hTR, suggesting that either increased triple helix formation or the absence of a terminal U sequence destabilizes the association of LARP3 with the exL form of hTR (Fig. 3b, lane 6). Analysis of in vitro telomerase assembly with the U460C mutant showed a decrease in the binding of LARP3 compared to the binding on wild-type hTR (Fig. 3c, lanes 7–11). The triple helix structure of the exL form plays a role in transiently protecting it from rapid degradation, which promotes the biogenesis process that yields the mature form[3]. Consistent with previous observations[3], the U460C hTR mutant affected the dynamics of 3′-end processing (Fig. 3d, lanes 1–10 and Supplementary Fig. 4a). Incubation of the U460C mutant for 60 min led to the production of large quantities of the mature form (Fig. 3d, lane 9) compared to the amount of the mature form obtained from wild-type hTR (Fig. 3d, lane 4). However, the 375-377GGAdel mutant, which was expected to have an exposed terminal U sequence and disrupted triple-helix formation, pulled down more LARP3 (Fig. 3b, lane 14), and LARP3 continued to bind to the 375-377GGAdel mutant relatively longer than to the wild-type hTR (Fig. 3c, lanes 18–22), which blocked the conversion of exL to its mature form and resulted in its degradation (Fig. 3d, lanes 11–20 and Supplementary Fig. 4b). Taken together, these data suggest that LARP3 binding to the UUU stretch of exL hTR competes with the formation of tertiary structures in the exL form, determining the efficiency of mature hTR production.

## LARP3 plays a negative role in telomerase biogenesis
Our results suggest a role for tertiary RNA interactions in the exL form of hTR, which promotes 3′-end processing via a mechanism that leads to competition between 3′-end processing and LARP3 binding (Fig. 3). We wondered whether LARP3 plays a negative role in hTR biogenesis. To determine the role of LARP3 in the processing of the exL form, we knocked down LARP3 in 293T cells using shRNAs targeting three different regions of the LARP3 coding sequence (Fig. 4a, lane 2 and Supplementary Fig. 5a, b). We found a minor effect on the steady-state hTR level in the LARP3 knockdown cells (Fig. 4b). LARP3 knockdown, however, did not substantially affect the assembly of DKC1 with hTR in vitro (Fig. 4d, lanes 1–12). However, telomerase activity increased by 20% when telomerase was purified by immunoprecipitation with DKC1 (Fig. 4e, lane 4). Subjecting LARP3 knockdown extracts to in vitro 3′ processing facilitated the biogenesis of hTR (Fig. 4f, lanes 7–12 and Supplementary Fig. 6a), in contrast to the results obtained with the control extracts (Fig. 4f, lanes 1–6). These results suggested that LARP3 plays a role in preventing 3′-end maturation.

To confirm the negative role of LARP3 in hTR 3′-end processing, we overexpressed LARP3 in 293T cells (Fig. 4a, lanes 3–4). LARP3 overexpression caused a 1.2-fold and 3.1-fold increase in all forms and in the 3′-exL form of hTR, respectively (Fig. 4c). In addition, a 40% decrease in telomerase activity was observed (Fig. 4e, lanes 5–8). An in vitro telomerase assembly assay revealed that although LARP3 had a minor effect

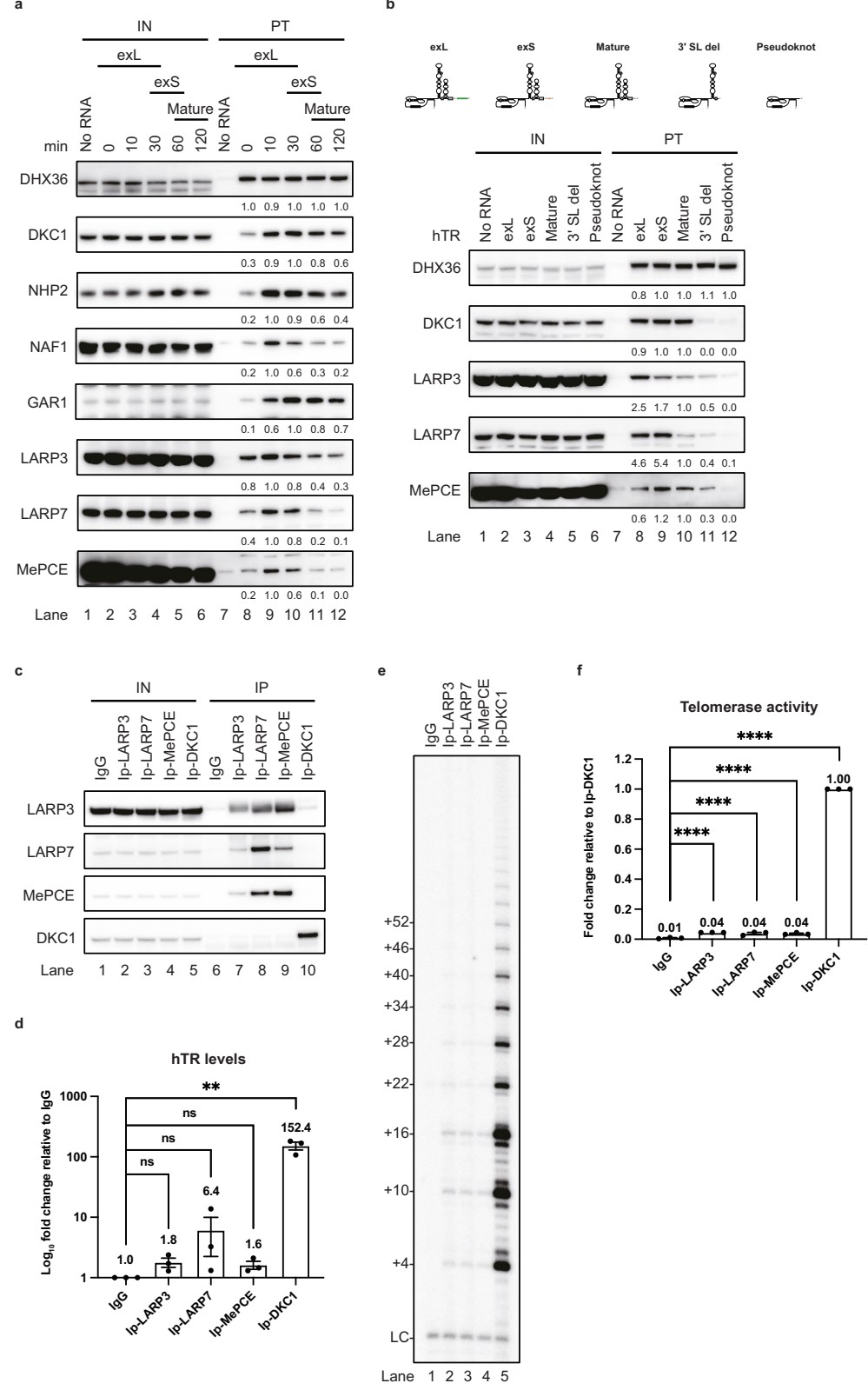

on DKC1 binding, LARP3 overexpression markedly blocked the binding of LARP7 and MePCE to hTR (Fig. 4d, lanes 20–24). Consistent with the observation that LARP3 overexpression increased the fraction of the exL form, the 3'-end processing of exL was profoundly blocked (Fig. 4g and Supplementary Fig. 6b). These data indicated that the binding of LARP3 to hTR prevented 3'-end processing of the exL form. In addition, the binding of LARP7/MePCE to hTR was blocked by LARP3 binding,

suggesting that a switch from LARP3 binding to LARP7/MePCE binding is required for telomerase RNA maturation.

## Reducing the expression level of LARP3 increases telomerase function and causes telomere elongation

Reducing the expression level of LARP3 in 293T cells could facilitate the biogenesis of hTR (Fig. 4f) and cause a subtle increase in

**Fig. 2 | LARP3, LARP7, and MePCE are involved in the early stage of telomerase assembly. a** Western blot analysis of in vitro-assembled telomerase purified at the indicated times. The signals from purified telomerase (lanes 7–12) were normalized to the peak signal. Input (IN) represents 10% of the purified telomerase (PT) samples. The samples derive from the same experiment and that gels/blots were processed in parallel. **b** Western blotting analysis of in vitro assembled telomerase assembled on the different hTR species as shown in the schematic (exL, exS, mature, 3′ stem loop-deleted, and pseudoknot). **c** Western blot analysis of 293T cell extracts after immunoprecipitation with antibodies against LARP3, LARP7, MePCE, and DKC1. **d** RT–qPCR quantification of hTR recovered from immunoprecipitations for LARP3, LARP7, MePCE, and DKC1 normalized to the IgG control. Bar graph of the mean fold change in the hTR relative to the control samples. The mean values +/−

SEM were calculated from triplicate qRT–PCR experiments of three biological replicates. Dots represent data points from individual experiments. The significance of the change in hTR between samples was calculated with a two-sided Student's t test; *$p < 0.05$, **$p < 0.01$, ***$p < 0.005$. **e** Endogenous LARP3, LARP7, MePCE, and DKC1 were immunoprecipitated and subjected to a telomerase activity assay. **f** Bar graph of the mean fold change in telomerase activity relative to that in the DKC1-associated telomerase samples. The mean values +/− SEM were calculated from triplicate telomerase activity experiments of three biological replicates. Dots represent data points from individual experiments. The significance of the change in telomerase activity between samples was calculated with a two-sided Student's t test; *$p < 0.05$, **$p < 0.01$, ***$p < 0.005$, ****$p < 0.001$. Source data are provided as a Source data file.

telomerase activity in 293T cells (Fig. 4e), although telomeres were not elongated (Supplementary Fig. 6c). Given that no reports have evaluated the levels of LARP3 in 293T cells and that the aberrant expression of LARP3 has been found in various cancers, including CML[22], and CML patients normally exhibit short telomeres[33]. This finding prompted us to speculate that reducing LARP3 expression in cell lines expressing high levels of LARP3 leads to increased telomere length. To evaluate this possibility, we depleted LARP3 in K562 cells (Fig. 5a and Supplementary Fig. 7a). The levels of both the mature and 3′-extended forms of hTR were increased in the LARP3 knockdown cells (Fig. 5b). Consistent with this observation, the in vitro 3′-end processing assay indicated that relatively more of the exL form was converted into mature hTR (Fig. 5c and Supplementary Fig. 7b). In addition, more telomerase was formed in vitro, as determined by more DKC1 molecules binding to hTR (Fig. 5d, lanes 8–10). Consistent with these findings, increased telomerase activity was observed (Fig. 5e). We measured telomere length by a telomere restriction fragment (TRF) assay. Telomeres were elongated in the LARP3 knockdown cells (Fig. 5f).

Overall, our data suggest that LARP3 plays a negative role in controlling telomere length by affecting telomerase biogenesis. Reducing the expression of LARP3 in cells with elevated levels of LARP3 facilitates the maturation of telomerase and boosts telomerase activity, which leads to telomere elongation.

### LARP7 and MePCE knockdown impairs the processing of the exS form and causes cytoplasmic localization of hTR

LARP7 and MePCE are mainly associated with the exS form in a step in which PARN is involved. However, how LARP7 and MePCE affect telomere homeostasis remains unclear. To evaluate the requirements of LARP7 and MePCE for telomerase maintenance, we knocked down LARP7 and MePCE in HeLa cells (Fig. 6a and Supplementary Fig. 8a) and 293T cells (Supplementary Fig. 9a). We measured telomere length in the LARP7 and MePCE knockdown HeLa (Fig. 6b) and 293T cells (Supplementary Fig. 9b). The LARP7 and MePCE knockdown cells had shorter telomeres than did the control cells (Fig. 6b and Supplementary Fig. 9b) and exhibited reduced telomerase activity (Fig. 6c, d and Supplementary Fig. 8b). Next, we examined the effects of LARP7 and MePCE knockdown on hTR levels (Fig. 6d and Supplementary Fig. 8c). Northern blots revealed reductions in the steady-state level of hTR upon knockdown of LARP7 and MePCE, although the effects were modest but highly reproducible (Fig. 6e and Supplementary Fig. 8c).

Since the resolution afforded by the 4% polyacrylamide gels did not allow us to distinguish between 451-nt mature hTR and slightly longer forms, we examined how the production of the 451-nt mature form is affected upon the knockdown of LARP7 and MePCE by RNA ligase-mediated rapid amplification of cDNA ends (RLM-RACE) coupled with high-throughput sequencing (Fig. 6f). Knockdown of LARP7 and MePCE resulted in an increase in the fraction of longer hTR transcripts with 3′ termini mapping beyond position 451 (Fig. 6f). To

support this observation, we performed an in vitro telomerase assembly assay. The results showed that MePCE bound to the exS form of hTR in the absence of LARP7 (Supplementary Fig. 9c, lanes 5–8) and vice versa (Supplementary Fig. 9c, lanes 9–12). In the absence of LARP7 or MePCE, LARP3 bound to the hTR relatively longer than it bound to the shRNA-treated control cell extracts (Supplementary Fig. 9c). We investigated the effect of LARP7 and MePCE knockdown on the processing of the exS form. Although exS was processed into the mature hTR in the extracts from the cells with LARP7 and MePCE knockdown (Supplementary Fig. 9d), the rate of conversion of the exS form to the mature form was lower in the extracts from the cells with both LARP7 and MePCE knockdown (Supplementary Fig. 9d, lanes 9 and 14) than in the control extracts (Supplementary Fig. 9d, lane 4). These data suggest that LARP7 and MePCE play a role in the conversion of the 3′-extended form of hTR to the mature form.

LARP7 and MePCE knockdown mimicked the effect of inhibited PARN activity on exS form processing[3], which substantially impaired the conversion of 3′-extended hTR to the mature form in vivo (Fig. 6f) and in vitro (Supplementary Fig. 9d). Cytoplasmic localization of hTR has been previously observed in PARN knockdown cells[40]. Induced pluripotent stem (iPS) cells from patients with PARN mutations produced short telomeres[8]. Consistent with these observations, PARN knockdown cells produced shorter telomeres (Fig. 6b, lanes 3 and 4), and 50% of hTR localized to the cytoplasm in the PARN knockdown cells, which was greater than that in the control cells (32%) (Fig. 6g, h). Although increased fractions of cytoplasmic hTR were observed upon LARP7 (45%) and MePCE (38%) knockdown, none of these differences were statistically significant (Fig. 6g, h). To support the findings of the imaging experiments, cell fractionation experiments revealed increases in the fractions of cytoplasmic hTR upon PARN, LARP7, and MePCE knockdown (Fig. 6i). Taken together, these results indicated that LARP7 and MePCE are involved in the conversion of the exS form to the mature form. Defects in the LARP7 and MePCE proteins caused the accumulation of hTR in the cytoplasm, reduced telomerase activity, and telomere shortening.

## Discussion

Human telomerase biogenesis is a highly dynamic process that is initiated by the precise stepwise binding of protein components to the RNA subunit hTR/TERC. Each step serves as a checkpoint for quality control and plays a decisive role in the production of mature telomerase versus the elimination of improper products[3,5]. A deficiency in telomerase components, including proteins and RNA, leads to degenerative human disease. Therefore, understanding telomerase biogenesis is critical for determining its medical relevance and elucidating the cause of telomere disorder syndrome. Unfortunately, the low abundance of endogenous telomerase precomplexes in cells makes it difficult to characterize the molecular mechanisms involved in vivo. To overcome these limitations, we established in vitro cell-free systems that allowed us to investigate telomerase assembly and 3′-end processing of hTR. Using these

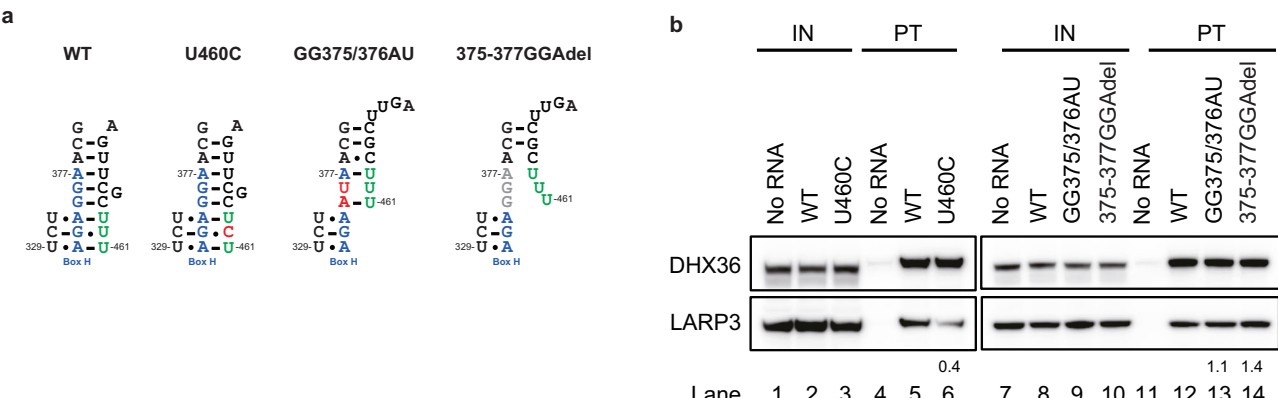

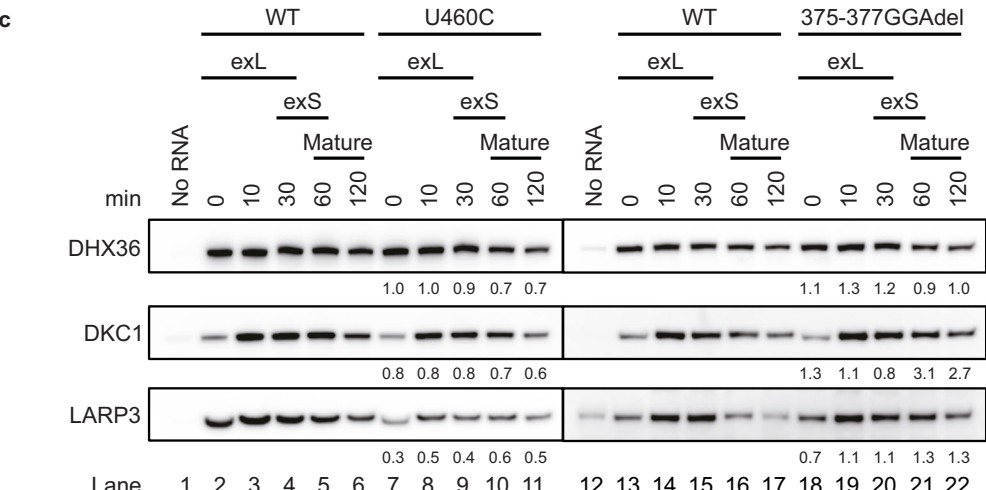

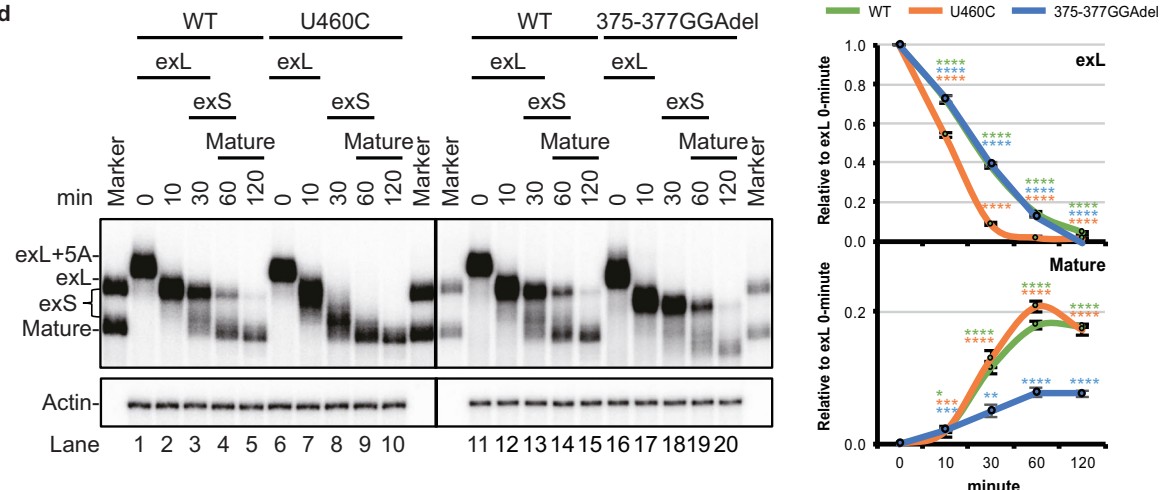

**Fig. 3 | LARP3 binding competes with tertiary structure formation. a** Schematic showing the proposed tertiary structure of exL with or without a mutation. **b** Western blot analysis of DHX36 and LARP3 pulled down with biotinylated wild-type, U460C, GG375/6AU, or 375-377GGA-deleted mutant hTR. **c** Western blotting analysis of telomerase assembled on biotin-labelled wild-type, U460C, or 375-377GGA-deleted mutant hTR pulled down with streptavidin beads for the indicated times. **d** The in vitro 3′ end processing assay with $^{32}$P-labelled wild-type, U460C, or

375-377GGA-deleted mutant hTR fragments was carried out in 293T cell extracts at 37 °C for the indicated times. RNA was purified and resolved on a 6% polyacrylamide gel containing 8 M urea. Actin served as the loading control. Data are presented as mean values +/− SD calculated from triplicate experiments of three technical replicates. Dots represent data points from individual experiments. The significance was calculated with a two-sided Student's t test; *$p < 0.05$, **$p < 0.01$, ***$p < 0.005$, ****$p < 0.001$. Source data are provided as a Source data file.

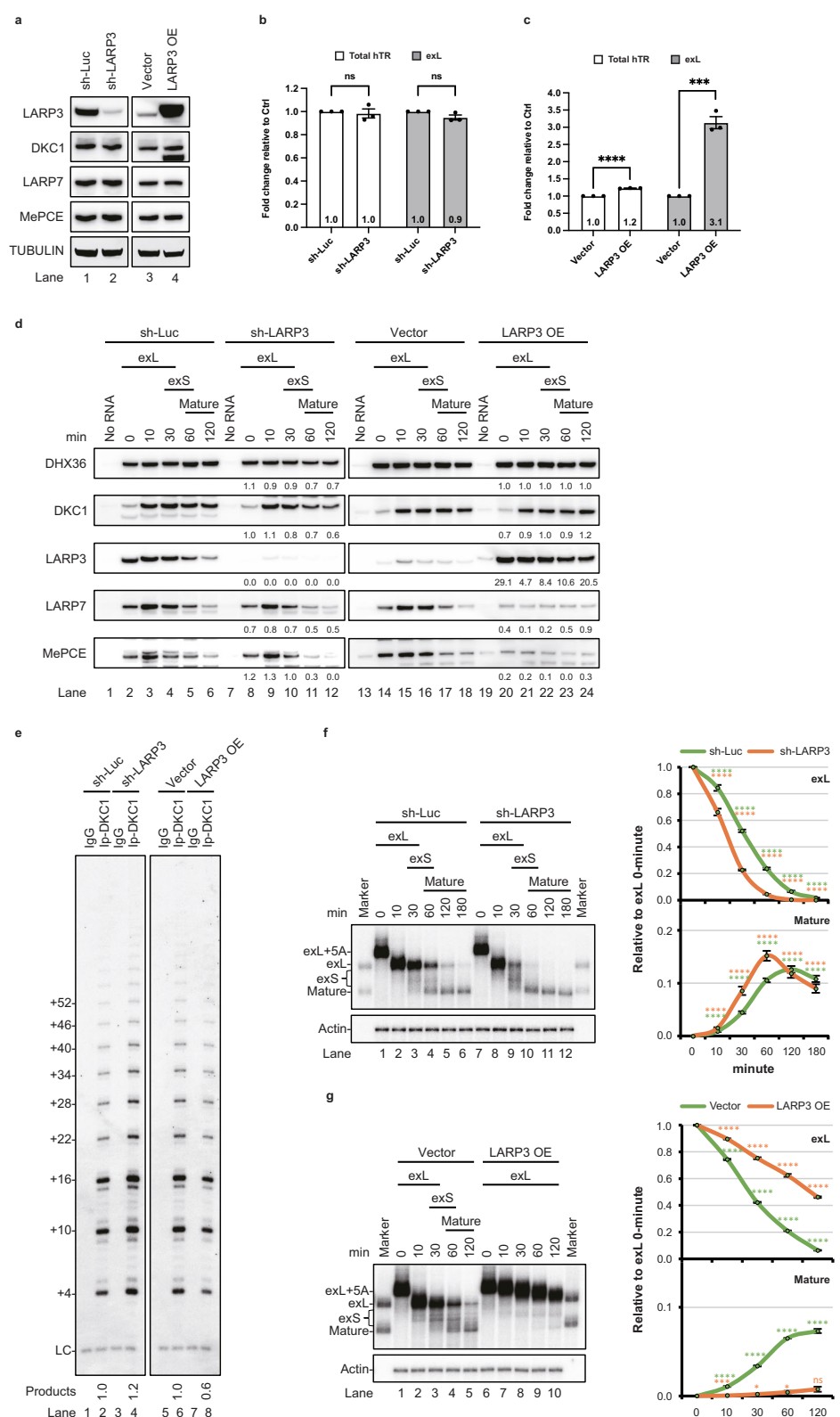

systems, we identified LARP3, LARP7, and MePCE as previously unknown players that are sequentially involved in the early stage of hTR biogenesis (Fig. 7).

LARP3 has been shown to bind with high affinity to 3′ uridylate residues of RNA polymerase III transcripts immediately upon transcription termination; these precursors include 5 S rRNA[41], tRNA[41] and 7SK RNA[42,43]. LARP3 acts as an RNA chaperone to prevent pre-tRNA

misfolding[44,45]. Similarly, our data indicated that LARP3 preferentially bound to the hTR precursor form exL concurrently with LARP7 and MePCE (Fig. 2). The association of LARP3 with the exL form was stabilized by a terminal U stretch (Fig. 3). LARP3 knockdown facilitated hTR maturation and telomerase activity (Fig. 4). In contrast, LARP3 overexpression clearly prevented both the processing and degradation of the exL form in vitro and reduced telomerase activity (Fig. 4).

**Fig. 4 | LARP3 plays a negative role in telomerase biogenesis. a** Western blots of cell extracts prepared from 293 T cells treated with either shRNA targeting LARP3 or transfected with an LARP3 plasmid. Endogenous tubulin served as a loading control. **b, c** Total RNA from 293 T cells treated with either shRNA targeting LARP3 (**b**) or transfected with an LARP3 plasmid (**c**) was subjected to qRT–PCR for total hTR, 3′-extended hTR, GAPDH, ATP5β, and HPRT. Bar graph of the mean fold change in the hTR level relative to that of the control samples normalized to that of GAPDH, ATP5β, and HPRT. The mean values +/− SEMA were calculated from triplicate qRT–PCR experiments of three biological replicates. Dots represent data points from individual experiments. The significance of changes between samples was calculated with a two-sided Student's t test; *$p < 0.05$, **$p < 0.01$, ***$p < 0.005$, ****$p < 0.001$. **d** Western blotting analysis of telomerase assembled on biotin-labelled hTR in the indicated extracts, followed by pulldown with streptavidin

beads for the indicated times. **e** LARP3 was immunoprecipitated from cell extracts prepared from 293T cells either treated with shRNA targeting LARP3 or transfected with an LARP3 plasmid and subjected to a telomerase activity assay. **f, g** The in vitro 3′ end processing assay with $^{32}$P-labelled hTR fragments was carried out in cell extracts prepared from 293T cells either treated with either shRNA targeting LARP3 (**f**) or transfected with an LARP3 plasmid (**g**) at 37 °C for the indicated times. RNA was resolved on a 6% polyacrylamide gel containing 8 M urea. Actin served as the loading control. The mean values +/− SD were calculated from triplicate experiments of three technical replicates. Dots represent data points from individual experiments. The significance was calculated with a two-sided Student's t test; *$p < 0.05$, **$p < 0.01$, ***$p < 0.005$, ****$p < 0.001$. Source data are provided as a Source data file.

Interestingly, the terminal U stretch is also essential for tertiary structure conformations in the exL form, protecting it from rapid degradation and creating an opportunity for hTR maturation[3]. The biogenesis pathway was highly activated when the tertiary structure of the exL isoform was stabilized after the introduction of a U460C mutation that attenuated LARP3 binding (Fig. 3). In contrast, LARP3 retained its ability to bind to the disease-related 375-377GGAdel mutant (Fig. 3). In this regard, the tertiary structure of the exL form is disrupted. These data indicate that the amount of mature hTR is determined by kinetic competition between LARP3 binding to the exL form and the formation of the tertiary exL structure. LARP3 recognizes the exL hTR that fails to fold into the triple helix, suggesting a role for LARP3 in quality control, which prevents the biogenesis of the faulty hTR (Fig. 7).

Our data suggest that LARP3 negatively regulates telomerase activity and telomere length at the level of telomerase biogenesis. Excessive LARP3 blocked the maturation of hTR, and the disassociation of LARP3 was essential for the assembly of functional RNPs with hTR. The average telomere length is normally shorter in CML patients than in healthy individuals[33]. Additionally, CML patients with long telomeres have been suggested to have a lower clinical risk profile than patients with short telomeres[46]. A study of the correlation between telomere length and CML progression suggested that patients in later phases (the accelerated phase and blast phase) present with considerably shorter telomeres than patients in the early phase (chronic phase)[33]. Notably, LARP3 expression is correlated with poor clinical prognosis in CML patients and increases during CML progression[32]. Our studies indicated that reducing the expression level of LARP3 in a CML cell line increased telomere length by promoting telomerase biogenesis and established a link between LARP3 expression and human telomerase biogenesis (Fig. 5).

A previous study suggested that RRP6 processes the terminal U tract and irreversibly disrupts the triple helix[3]. DKC1 subsequently establishes the 451-nt end by attenuating the 3′-end processing of the exS form via PARN, suggesting that structural rearrangements of hTR are required for efficient maturation[3]. Our data indicated that the compositional exchange of protein components occurs during this process (Fig. 7). NAF1 is replaced with GAR1. LARP3 appears to disassociate from the exL form of hTR. The mutually exclusive interaction of 7SK RNP with LARP3 or LARP7 has been suggested, and LARP3 needs to be replaced by LARP7 for the maturation of 7SK RNPs[42]. This may be the case with hTR. The binding of LARP7 and MePCE to hTR was attenuated by LARP3 overexpression (Fig. 4d). Notably, LARP3 maintained the association with the exS form longer in the absence of LARP7 and MePCE (Supplementary Fig. 9c), suggesting that LARP7 and MePCE bind instead of LARP3. We found that LARP7 and MePCE knockdown impaired the PARN-mediated processing of the exS form into the mature form and, similar to PARN knockdown, caused cytoplasmic accumulation of hTR. Together with these observations, these data support a model in which LARP7 and MePCE promote the transition of a LARP3-associated pre-telomerase to an H/ACA complex-

associated telomerase that promotes the conversion of the exS form into the mature form. Once this conversion is impaired, hTR is either degraded by the RNA exosome in the nucleus or exported to the cytoplasm and degraded by DCP2-XRN1.

Loss of function of *LARP7* and *MePCE* has been shown to cause Alazami syndrome[21] and neurodevelopmental disorders[29], respectively. Reduced expression of LARP7 has been shown to reduce telomerase activity and result in progressively shorter telomeres in human cancer cell lines[21]. Previous works with *S. pombe* demonstrated that Pof8 plays a key role in quality control of *S. pombe* telomerase RNA folding and forms a complex with Bmc1, the orthologue of MePCE, and Telomerase Holoenzyme Component 1 (Thc1), which promotes the assembly of a functional telomerase[26–28,30,31,47]. Like that associated with *S. pombe* Bmc1, telomere shortening has been observed in MePCE-deficient human cells. Interestingly, Thc1 shares structural similarity with the nuclear cap-binding complex and PARN[30]. LARP7 and MePCE knockdown impaired the conversion of the exS form to the mature form (Fig. 6f and Supplementary Fig. 9d). LARP7 bound hTR in a MePCE-independent manner and was required to stabilize the interaction of MePCE with hTR (Supplementary Fig. 9c). Although LARP7 and MePCE contribute to the conversion of the exS form to the mature form, they do not remain stably associated with active telomerase (Fig. 7). Compared to *S. pombe* telomerase RNA, an additional 3′-end processing step is required for the 3′ end maturation of telomerase RNA after spliceosomal cleavage in other species[48,49]. An investigation into the requirement of Pof8 for 3′-end processing in these species may yield interesting results. Our data not only suggest an evolutionary link in the biogenesis of telomerase among distant organisms but also provide new insights into the mechanisms underlying the pathogenesis of LARP3 and LARP7/MePCE deficiencies.

## Methods
### Preparation of hTR RNA substrates
In vitro transcription reactions were carried out in 1X transcription buffer C (Croyez), 0.5 mM ATP (Croyez), CTP (Croyez), and GTP (Croyez); 0.1 mM UTP (Croyez); α-$^{32}$P-UTP (3000 Ci mmol$^{-1}$, 10 mCi ml$^{-1}$, PerkinElmer); 0.2 μg of DNA template; 40 U of RNasin (TOOLs); and 1 unit μl$^{-1}$ T7 RNA polymerase (Croyez). The reaction mixtures (10 μl) were incubated at 37 °C for 30 min, followed by the addition of an equal volume of formamide dye. The RNA products were purified on a 6% polyacrylamide (19:1) gel containing 8 M urea. The primers used to generate the DNA templates are listed in Supplementary Table 1. The loading control actin-1 RNA was expressed via Sp6 RNA polymerase (Promega).

### Cell extract preparation
Human cell extracts were prepared from cell pellets ($2 \times 10^7$ cells) with 250 μl of CHAPS buffer (0.5% CHAPS; 50 mM Tris-HCl, pH 8.0; 50 mM KCl; 1 mM MgCl$_2$; 1 mM EGTA; 10% glycerol; 5 mM DTT; 1 mM PMSF). The cells were lysed at 4 °C on a nutator for 1 h. The lysate was then centrifuged at $21,130 \times g$ at 4 °C for 10 min. The supernatant was

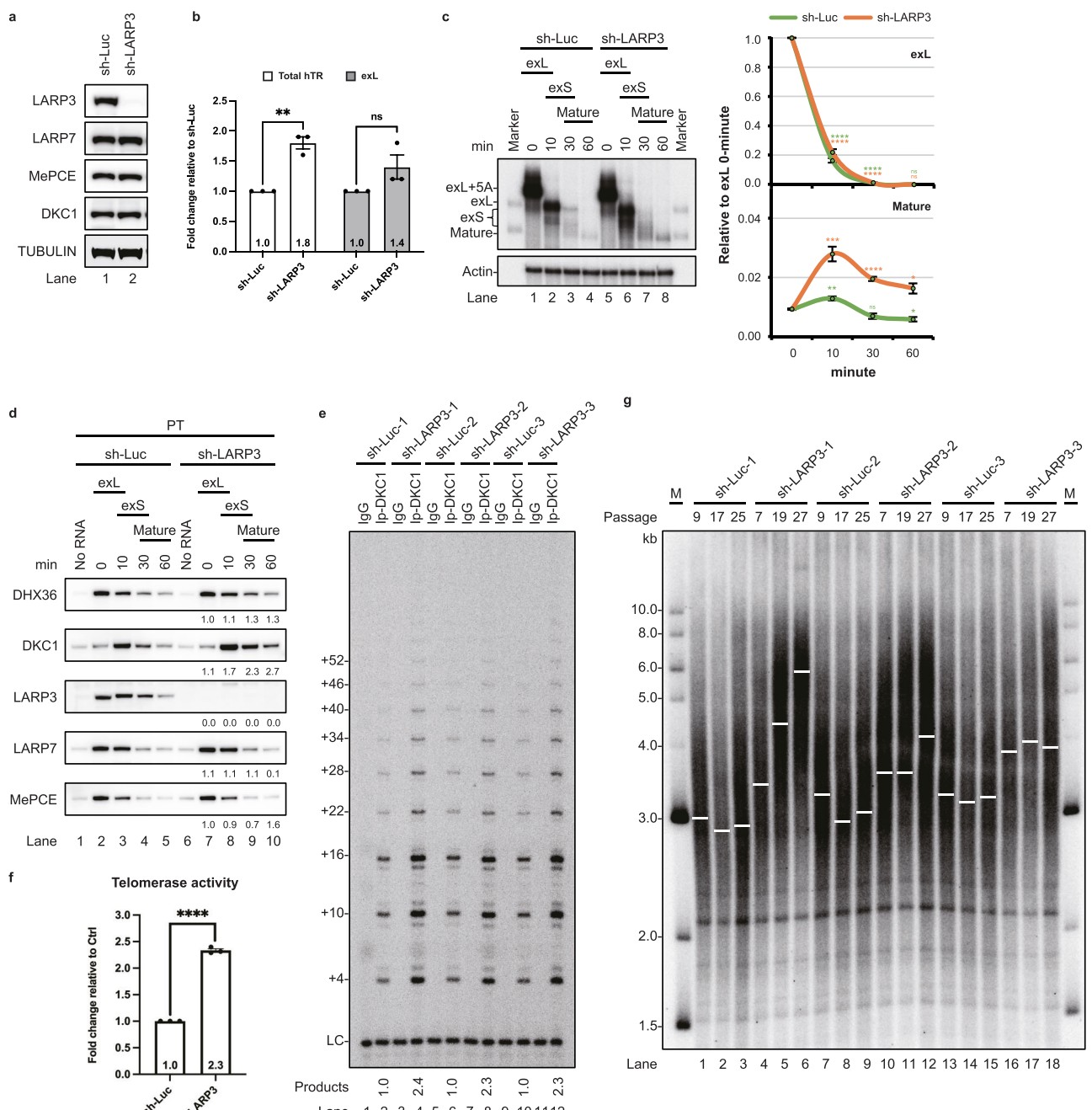

**Fig. 5 | Reducing the expression level of LARP3 increases telomerase function and causes telomere elongation. a** Western blots of cell extracts prepared from K562 cells treated with shRNAs targeting luciferase or LARP3. Endogenous tubulin served as a loading control. **b** Total RNA from LARP3 knockdown K562 cells was subjected to qRT–PCR to measure the levels of total hTR, 3'-extended hTR, GAPDH, ATP5β, and HPRT. Bar graph of the mean fold change for 3'-extended hTR relative to that of the control samples normalized to GAPDH, ATP5β, and HPRT. The mean values +/− SEM were calculated from triplicate qRT–PCR experiments of three biological replicates. Dots represent data points from individual experiments. The significance of changes between samples was calculated with a two-sided Student's t test; *p < 0.05, **p < 0.01, ***p < 0.005, ****p < 0.001. **c** An in vitro 3' end processing assay with ³²P-labelled hTR fragments was carried out in the indicated cell extracts. RNA was resolved on a 6% polyacrylamide gel containing 8 M urea. Actin served as the loading control. The mean values +/− SD were calculated from triplicate experiments of three technical replicates. Dots represent data points from individual experiments. The significance was calculated with a two-sided Student's t test; *p < 0.05, **p < 0.01, ***p < 0.005, ****p < 0.001. **d** Western blotting analysis of telomerase assembled on biotin-labelled hTR in the indicated extracts, followed by pulldown with streptavidin beads for the indicated times. **e** Endogenous DKC1 was immunoprecipitated and subjected to a telomerase activity assay. **f** Bar graph of the mean fold change in telomerase activity relative to the control group. The mean values +/− SEM were calculated from three biological replicates with a two-sided Student's t test. **g** Telomere lengths determined by TRF analysis of gDNA prepared from K562 cells treated with shRNAs targeting luciferase or LARP3. Source data are provided as a Source data file.

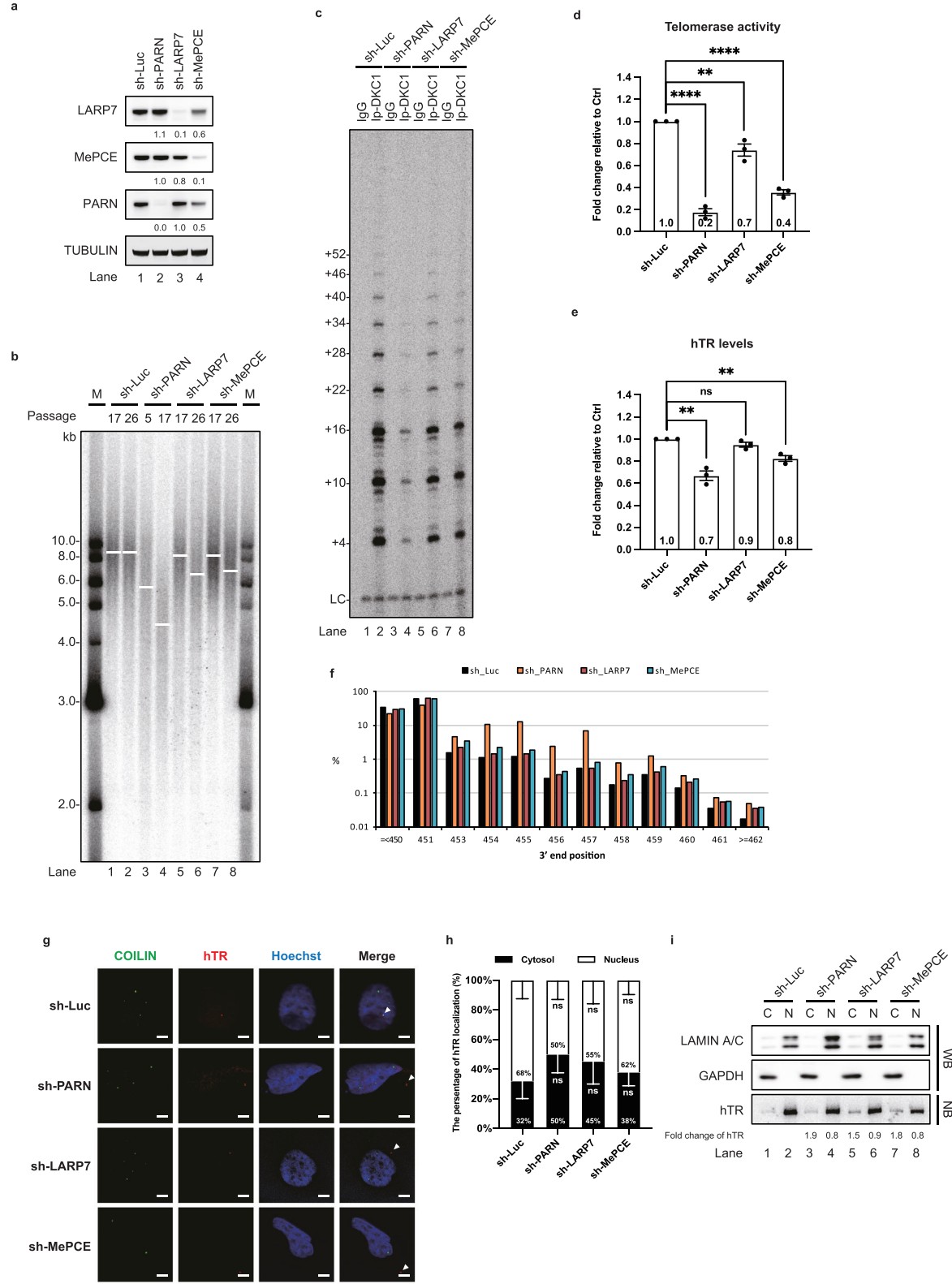

transferred to a new tube and centrifuged at 21,130 × *g* at 4 °C for another 10 min. The protein concentrations were measured by Bradford assay (Bio-Rad). The cell extracts were stored at −80 °C.

**In vitro hTR 3′-end processing assay**
In vitro hTR processing reactions (10 μl) were carried out at 37 °C in buffer containing 20 mM Tris-HCl (pH 7.5), 50 mM KCl, 2.5 mM MgCl₂,

40 U of murine RNase inhibitor (Croyez), 2 nM ³²P-labelled hTR RNA, and 40 μg of whole-cell extracts. Reactions were stopped by the addition of stop buffer (10 mg ml⁻¹ proteinase K in 0.5% SDS; 40 mM EDTA; 20 mM Tris-HCl, pH 7.5; and 1000 c.p.m. μl⁻¹ ³²P-labelled actin mRNA) and incubation at 37 °C for 30 min, followed by extraction with phenol/chloroform preequilibrated with 50 mM NaOAc (pH 5.0) and ethanol precipitation. The RNA pellet was dissolved in 80% formamide

**Fig. 6 | LARP7 and MePCE knockdown impairs the hTR processing and cellular localization. a** Western blots of cell extracts prepared from the indicated shRNA-treated HeLa cells. **b** TRF analysis of gDNA prepared from the indicated shRNA-treated HeLa cells. **c** Telomerase activity assay of endogenous DKC1 immunoprecipitates from the indicated shRNA-treated HeLa cell extracts. **d** Bar graph of the mean fold change in telomerase activity relative to the control group. The mean values +/− SEM were calculated from three biological replicates with a two-sided Student's t test. **e** Northern blots of total RNA prepared from the indicated shRNA-treated HeLa cells. Bar graph of the mean fold change in hTR levels relative to the control and normalized to U1 snRNA levels. The mean values +/− SEM were calculated from three biological replicates with a two-sided Student's t test. **f** Bar graph showing the distribution of hTR 3' end positions mapped using 3' RACE present on a subset of the transcripts. The numbers of reads analysed were as follows: sh-luciferase, 5,827,715; sh-PARN, 4,725,159; sh-LARP7, 5,008,122; and sh-MePCE, 6,105,815. **g** In situ hybridization and IF in the indicated shRNA-treated HeLa cells. Coilin served as a Cajal body marker. The scale bar represents 5 μm. **h** Bar graph illustrating the distribution of hTR shown in (**g**) in the cytosolic and nuclear fractions. The mean values +/− SEM were calculated from three biological replicates with a two-sided Student's t test. **i** Total protein and RNA prepared from nuclear or cytosolic fractions from the indicated shRNA-treated HeLa cells were subjected to Western blots and Northern blots, respectively. Lamin A/C and GAPDH served as a nuclear marker and a cytosolic marker, respectively. The fold change in the hTR levels relative to those in the control samples was normalized to the levels of Lamin A/C or GAPDH, respectively. Dots shown in the bar graph represent data points from individual experiments. $p$ values; $*p < 0.05$, $**p < 0.01$, $***p < 0.005$, $****p < 0.001$. Source data are provided as a Source data file.

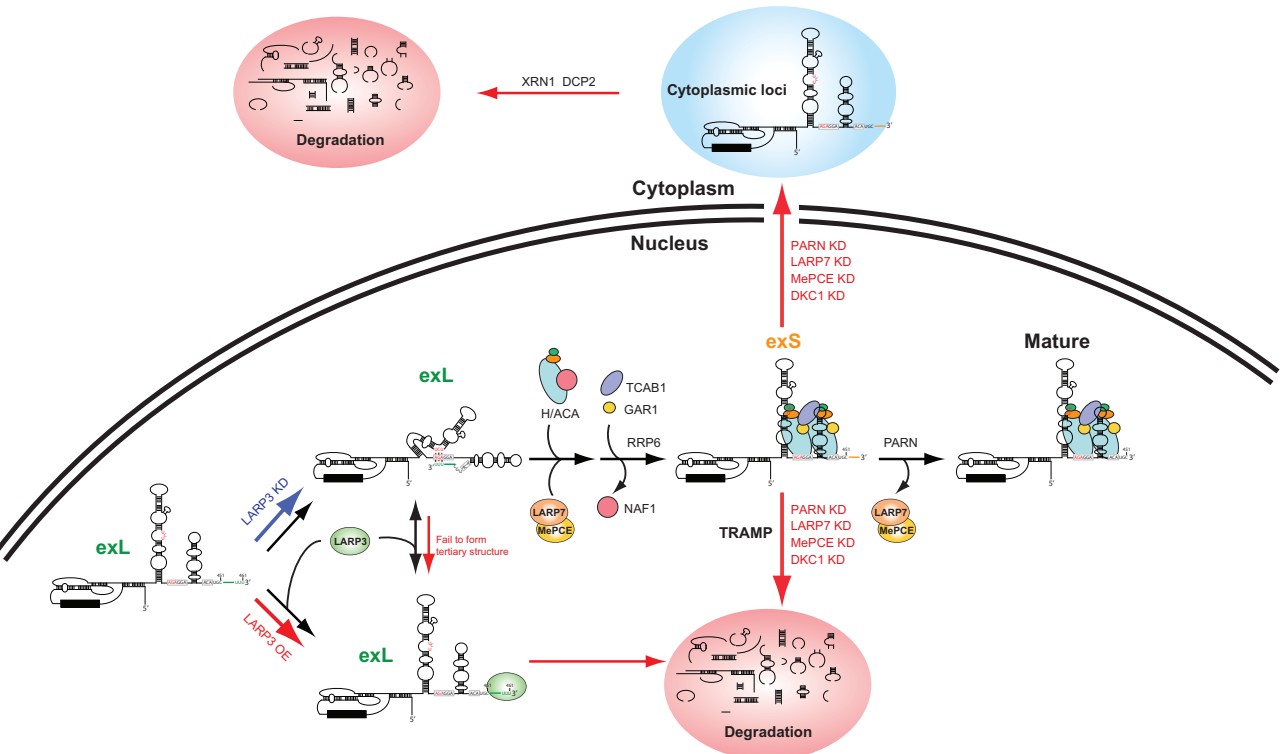

**Fig. 7 | The proposed model of the role of LARP3, LARP7, and MePCE during the stepwise assembly of human telomerase.** After transcription of human telomerase RNA (hTR), the 3'-extended long (exL) form without triple helix formation is preferentially recognized by LARP3 and is subsequently degraded. The triple helix conformation prevents LARP3 binding, which protects the exL form of hTR from rapid degradation and facilitates hTR biogenesis. PARN, LARP7, and MePCE function during the conversion of the 3'-extended short (exS) form into mature hTR. The absence of PARN, LARP7, and MePCE impairs the processing of exS and causes cytoplasmic localization of hTR.

dye and analysed on a 6% polyacrylamide (19:1) gel containing 8 M urea.

### Preparation of capped hTR RNA substrates

In vitro transcription reactions were performed in 1X transcription buffer (Promega), 0.5 mM nucleoside 5-triphosphates (NTPs), 25 μM Bio11 UTP, α-$^{32}$P-UTP (3000 Ci mmol$^{-1}$, 10 mCi ml$^{-1}$, PerkinElmer), 0.1 μg of DNA template, 40 U of murine RNase inhibitor (Croyez), and 1 U of SP6 RNA polymerase (RiboMAX™, Promega). The reaction mixtures (100 μl) were incubated at 37 °C for 4 h and then treated with DNase I (New England Biolabs) at 37 °C for 1 h, followed by extraction with phenol/chloroform preequilibrated with 50 mM NaOAc (pH 5.0) and ethanol precipitation. RNA was dissolved in 80% formamide dye and purified on a 4% polyacrylamide (29:1) gel containing 8 M urea. The capping reactions were carried out in 1X capping buffer (Croyez), 0.5 mM GTP, 0.1 mM SAM, and 1 U of vaccinia virus capping enzyme (Croyez). The reaction mixtures were incubated at 37 °C for 4 h, followed by extraction with phenol/chloroform preequilibrated with 50 mM NaOAc (pH 5.0) and ethanol precipitation. The RNA pellet was dissolved in ddH$_2$O. The RNA concentrations were measured by a NanoDrop Spectrophotometer.

### Telomerase pulldown

Telomerase was assembled in buffer containing 20 mM Tris-HCl (pH 7.5), 50 mM KCl, 2.5 mM MgCl$_2$, 40 U RNasin, 50 nM capped hTR RNA, and 10 μg of whole-cell extracts. The reaction mixture (25 μl) was incubated at 37 °C for the desired times, followed by centrifugation at 21,130 × $g$ at 4 °C for 2 min. The supernatant was incubated with the streptavidin beads at 4 °C for 1 h. The precipitates were washed with NET-2 buffer (50 mM Tris-HCl pH 7.5, 150 mM NaCl, and 0.05%

NP-40) and then subjected to Western blotting or telomerase activity assays.

## Immunoblotting

Cell extracts were diluted in 1×LDS sample buffer. Proteins in cell lysates were loaded onto a 4–20% Tris–glycine protein gel (mPAGE™ 4–20% Bis-Tris, Millipore) and transferred to a PVDF blot membrane (Bio-Rad). Low-fat milk (5%) in wash buffer (10 mM Tris-HCl, pH 8.0; 150 mM NaCl; 1 mM EDTA; and 10% Triton X-100) was used as a blocking reagent. The antibodies used in this study are listed in Supplementary Table 3.

## Immunoprecipitation

Ten micrograms of antibodies (IgG, LARP3, LARP7, MePCE, and DKC1) were conjugated to 10 µl of Protein A Sepharose™ CL-4B beads (Cytiva). The antibody-conjugated beads were incubated with 1 mg of the 293T cell extracts at 4 °C for 1 h. Protein A Sepharose™ CL-4B beads were washed with NET-2 buffer (50 mM Tris-HCl, pH 7.5; 150 mM NaCl; 0.05% NP-40) twice at $21,130 \times g$ for 10 seconds at 4 °C, followed by Western blotting, analysis of telomerase activity, and RT–qPCR. The antibodies used in this study are listed in Supplementary Table 3.

## Cell culture and transduction

293T cells (ATCC® CRL-3216™) were maintained in DMEM (Gibco) supplemented with 10% heat-inactivated foetal bovine serum (Corning) and 2 mM L-glutamine (Gibco) at 37 °C in a humidified atmosphere containing 5% $CO_2$. HeLa cells (ATCC® CCL-2™) were maintained in DMEM (Gibco) supplemented with 10% heat-inactivated foetal bovine serum. K562 cells (Horizon, HD PAR-131) were maintained in IMEM (HyClone™) supplemented with 10% heat-inactivated foetal bovine serum. The cells were subcultured when the confluency reached 80%. The cells were transfected with 15 µg of plasmid DNA using *Trans*IT® LT1 (Mirus) for 24 h. The plasmids used for transfection are listed in Supplementary Table 2. Cells were transduced with shRNAs for 24 h. Medium containing 2 µg ml⁻¹ puromycin was used to select the knockdown cells. Information on the shRNAs used for transduction is presented in Supplementary Table 4.

## RNA extraction and reverse transcription PCR

Total RNA was isolated from the pellets ($1 \times 10^7$ cells) using Ambion TRIzol® Reagent (Life Technologies, Cat. No: 15596018) according to the manufacturer's instructions, followed by DNase I (New England Biolabs) treatment at 37 °C for 60 min. For cDNA synthesis, reverse transcription was performed according to the manufacturer's instructions for the SuperScript™ IV First-Strand Synthesis System (Thermo Fisher Scientific, Cat. No. 18091050). In brief, 2 µg of total RNA was mixed with 50 µM Oligo d(T)20 primer or 50 ng of random hexamers in 10 mM dNTP mix. Reactions were carried out in a T100™ Thermal Cycler (Bio-Rad) under the following conditions: one cycle at 65 °C for 5 min and 4 °C for 1 min, followed by the addition of a reaction mixture containing 5 mM DTT, 1X SSIV buffer, murine RNase inhibitor (Croyez), and 1 µl of SuperScript™ IV RT. The reaction mixture was incubated at 50 °C for 1 h, 80 °C for 10 min, 37 °C for 1 min and 4 °C for 1 min. Then, 1 µl of RNase H (2 U/µl) was added to the reaction and incubated at 37 °C for 20 min, 65 °C for 10 min, and 4 °C for 10 min.

## Genomic DNA extraction

Genomic DNA was prepared from pellets ($5 \times 10^6$ cells) with a GenElute™ Mammalian Genomic DNA Miniprep Kit (Sigma-Aldrich, Cat. No: G1N350-1KT) according to the manufacturer's instructions.

## Northern blotting

Total RNA (10 µg) was separated on a 4% polyacrylamide (29:1) gel containing 8 M urea at 20 W for 1 h and then transferred to a Hybond-N + nylon transfer membrane (GE Healthcare) at 400 mA for 1 h in 0.5X TBE buffer. RNA was subsequently crosslinked twice to the membrane at 120 mJ in a UV Stratalinker 1800 (Stratagene, 254 nm, 120 mJ). The blot was prehybridized in Church buffer at 65 °C for 1 h and hybridized with $^{32}$P-dCTP-labelled hTR overnight. The probes used in this study are listed in Supplementary Table 5.

## Terminal restriction fragment analysis

Genomic DNA (1 µg) from 293T cells was digested with HinfI (New England Biolabs) and RsaI (New England Biolabs) restriction enzymes in 10X CutSmart® Buffer (NEB) at 37 °C overnight. The digested gDNA fragments were separated on a 1% SeaKem® LE agarose gel (Lonza) by electrophoresis at 120 V for 12 h, followed by capillary transfer to a Hybond-N⁺ nylon transfer membrane (GE Healthcare) in 10× saline sodium citrate (SSC) for 14 h. DNA was subsequently crosslinked twice to the membrane at 120 mJ in a UV Stratalinker 1800 (Stratagene, 254 nm, 120 mJ). The blot was prehybridized in Church buffer at 65 °C for 1 h and then hybridized with $^{32}$P-dCTP-labelled $(TTAGGG)_3$ overnight. The blot was exposed to a phosphor imaging screen (Fujifilm) at room temperature overnight. Phosphor images were scanned by an Amersham Typhoon 5 scanner (Cytiva). The telomere length images were quantified and analysed by ImageQuantTL software (Cytiva).

## Telomerase activity assay

Telomerase activity reactions were performed in a 10-µl reaction volume consisting of 50 mM Tris-HCl, pH 8.0; 50 mM KCl; 1 mM MgCl₂; 1 mM spermidine; 5 mM DTT; 1 mM dATP; 1 mM dTTP; 10 µM dGTP; 0.75 µM $^{32}$P-α-dGTP (3000 Ci mmol⁻¹); 1 µM telomeric primer $(TTAGGG)_3$; and 2 µg of cell extract at 37 °C for 2 h. Reactions were stopped with 10 µl of 1 mg ml⁻¹ proteinase K. DNA was extracted with phenol/chloroform preequilibrated with 50 mM NaOAc (pH 7.0) and ethanol precipitated with 2.5 M ammonium acetate and 10 µg of glycogen at −80 °C overnight. The reactions were then centrifuged for 20 min at $18,407 \times g$. The pellets were washed with 1 ml of 70% ethanol. The dried pellets were then resuspended in 5 µl of 80% formamide loading buffer. The reaction products were analysed on a 10% polyacrylamide (19:1) gel containing 8 M urea at 80 W for 1 h and 40 min. The blot was dried and then exposed to a phosphor imaging screen (Fujifilm) at room temperature overnight. Phosphor images were scanned by an Amersham Typhoon 5 scanner (Cytiva). For telomerase activity measurement, the intensity of all bands was measured and normalized to that of the control. For telomerase processivity measurements, the intensity of each major repeat band was measured and normalized to both the intensity of the first band and the number of $^{32}$P-labelled nucleotides incorporated. Normalized intensities were then plotted versus the number of repeats.

## qRT–PCR

Quantitative reverse transcription (qRT)–PCR was performed via the SYBR Green method. The 50-fold diluted random hexamer-primed cDNA was amplified with the primers shown in Supplementary Table 6 via a CFX384™ Real-Time PCR System in a C1000 Touch™ Thermal Cycler (Bio-Rad) using iQ™ SYBR® Green Supermix (Bio-Rad, Cat. No. 1708882). The results were normalized to the GAPDH, ATP5β, and HPRT reference gene levels and measured by CFX Maestro software (Bio-Rad). Graphing and statistical analysis of the qRT–PCR results were performed using Prism 9 (GraphPad).

## In situ hybridization (FISH) and immunofluorescence (IF)

Cells were fixed on coverslips with 4% paraformaldehyde (Thermo Scientific, Cat. No. 047317) and permeabilized with 0.1% Triton

X-100. The cells were hybridized with hybridization buffer (2X SSC, 10% formamide, 0.2 mg ml$^{-1}$, 10% dextran sulfate, 0.4 U RNase inhibitor, 1 mg ml$^{-1}$ *Escherichia coli* tRNA). The cells were incubated with seven Cy3-conjugated hTR oligos (Supplementary Table 7) at 37 °C overnight. For IF experiments, cells were incubated with an anti-Coilin primary antibody (Abcam, Cat. No. ab11822, 0.5 μg ml$^{-1}$) in 1% bovine serum albumin (BSA) for 2 h, followed by incubation with a FITC-conjugated AffiniPure goat anti-mouse IgG (H + L) (Jackson ImmunoResearch, Cat. No. 115-095-003, 1:100 dilution) secondary antibody for 1 h. Cells were stained with Hoechst 33258 (Sigma-Aldrich, Cat. No. B2883-1g, 1:1000 dilution) in 1% BSA for 10 min. Coverslips were mounted with Fluoromount™ Aqueous Mounting Medium (Sigma-Aldrich, Cat. No. F4680-25ML). The images were photographed with a Carl Zeiss LSM880 microscope.

### Site-directed mutagenesis

The plasmid pMG80, which contains a DNA fragment of wild-type hTR, was used as a template. The GG375/376AT or 375-377delGGA hTR mutation was introduced using a QuikChange® Lightning Mutagenesis Kit (Agilent) according to the manufacturer's instructions. The primers used in this study are listed in Supplementary Table 8.

### Cell fractionation

The cell subcompartments were separated with a Cell Fractionation Kit (Abcam, Cat. No: ab109719) according to the manufacturer's instructions. After fractionation, the cell lysates of the nucleus and cytosol were diluted in 5× SDS dye, subjected to Western blotting analysis or mixed with Ambion TRIzol® Reagent (Life Technologies) for RNA purification according to the manufacturer's instructions. The RNA concentrations were measured by a NanoDrop Spectrophotometer.

### 3′ RACE sequencing

The oligonucleotides used for sample preparation are listed in Supplementary Table 9. Ligation reactions contained 2.0 μg of total RNA, 1×T4 RNA ligase buffer, 5% PEG8000, 1 U of T4 RNA ligase I, and 5 μM 3′ linker (CKOligo-666). The ligation reaction mixture was incubated at 16 °C for 18 h and inactivated at 65 °C for 15 min. A total of 20 μl of reverse transcription reaction mixture containing 1X ligation reactions, 1× SSIV buffer, 0.1 μM RT primer (CKOligo-667), 0.5 μM dNTPs, 5 mM DTT, RNase inhibitor, and SuperScript™ IV Reverse Transcriptase (Invitrogen) was incubated at 50 °C for 10 min. A total of 10 U of *E. coli* RNase H was directly added to the reverse transcription reaction mixture, followed by incubation at 37 °C for 20 min and then heat inactivation at 70 °C for 15 min. The first round of PCR was performed with 0.5 μM CKOligo-667 and CKOligo-668 in 50 μl reactions containing 1× Phusion HF buffer, 0.2 mM dNTPs, and 0.02 U Phusion Hot Start II High Fidelity DNA polymerase (Thermo Scientific). First-round PCR products were purified with a PCR purification kit (TOOLS). For the second-round PCRs, 5 μl of first-round PCR product was used as a template. CKOligo-669 (0.5 μM) was used as the forward primer for all samples, and the reverse primers CKOligo-676 (sh-Luc), CKOligo-685 (sh-PARN), CKOligo-686 (sh-LARP7), and CKOligo-687 (sh-MePCE) were used. Amplicons were purified with a PCR purification kit (TOOLS). The samples were quantified by a Qubit fluorometer and a bioanalyzer, which were multiplexed and split between two lanes of a RapidSeq flow cytometer for sequencing. The library samples were sequenced on a NovaSeq X plus using RapidSeq-150 bp paired-end reads. A 10-nt molecular barcode was used to remove PCR duplicates. After quality filtering, 4.7 to 6.1 million reads were analysed per sample. A minimum match of 20 nts to the hTR reference sequence and 10 nts to the linker sequence was required for each read to pass the filter. For each filtered read, the most 3′ hTR reference coordinate between hTR:366 and 641 was identified by matching 20 nts closest to the 3′ end, allowing two mismatches not including the two most 3′ bases. Nucleotides found between the reference hTR and the linker sequence were considered nontemplated nucleotide additions (NTNAs). All reads were included in the 3′ end analysis irrespective of the presence or absence of NTNA.

### Statistics and reproducibility

For all bar/line graphs, two-tailed unpaired Student's t tests were performed. Significant values in the figures are marked (*$p < 0.05$, **$p < 0.01$, ***$p < 0.005$, ****$p < 0.001$). Experiments for Figs. 1a, 1d, 1e, 2e, 3b, 4e, 5a, 6a, 6c, and 6g were performed independently three times. Experiments for Figs. 1c, 2a, 5e, 5g, and 6b were performed twice. Experiments for Figs. 2b, 2c, 3c, 4a 4d, 5d, and 6i were performed once.

### Reporting summary

Further information on research design is available in the Nature Portfolio Reporting Summary linked to this article.

## Data availability

The primary sequence data associated with this analysis have been deposited under GEO accession number GSE262590. All reagents described in this study are available upon request from the corresponding author. Source data are provided with this paper.

## Code availability

The custom Python scripts used in this study are available at https://github.com/CK-Tseng-lab/3-RACE.git.

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

## Acknowledgements

We thank all members of the Tseng laboratory for helpful discussions. This work was supported by MOST 111-2636-B-002-026 and NTU-112V1403-5. We thank Dr. Chung-En Ni at Genomics, BioSci & Tech Co., Taiwan, for assisting with the 3' RACE analysis.

## Author contributions

T.-L.K., P.B. and C.-K.T. designed the experiments. T.-L.K. performed most of the experiments. Y.-H.C. performed the IF assays, FISH, and cell fractionation. T.-L.K and Y.-C.H. performed the Western blot, Southern blot, and Northern blot analyses. All the authors analysed the data. T.-L.K. and C.-K.T. wrote the manuscript.

## Competing interests

The authors declare no competing interests.
