## [Peer Review File · Nature Communications]

LARP3, LARP7, and MePCE are Involved in the Early Stage of Human Telomerase RNA BiogenesisREVIEWER COMMENTS

Reviewer #1 (Remarks to the Author):

In this manuscript, Kao and colleagues report that the La-related proteins LARP3, LARP7 and MePCE are involved in the early processing of the telomerase RNA hTR during telomerase biogenesis in human cells. Previous studies revealed that La-related proteins are involved in the 3' end processing of telomerase RNA in the fission yeast and ciliates. So far, no evidence for a role of LARP proteins in the biogenesis of human telomerase has been reported yet. This manuscript now shows that the human La-related proteins LARP3, LARP7 and MePCE are also involved in the regulation of hTR 3' end processing, control the assembly of telomerase holoenzyme and impact telomeres length homeostasis.

However, this manuscript falls short of providing convincing experimental evidence to support the claims made by the authors. Several controls are missing and some results are contradictory (see major comments below). As it is, this manuscript is not acceptable for publication in this journal, and would require major modifications in order to be publishable.

Major comments

Figure 1A: To what extent this assay faithfully reproduces the 3' end processing of hTR in vivo? The authors do not provide evidence that their in vitro hTR maturation assay depends on the exosome and the exonucleases PARN and TOE1. The authors should validate their assay by depleting exosome components, PARN and TOE1 using RNAi. Furthermore, it is not clear that exL RNA processing in this in vitro assay occurs exclusively from the 3' end. Is it possible that some of the degradation occurs from the 5' end?

Figure 1B: The majority (80%) of exL RNA is degraded in the in vitro assay, and only 20% is converted to the mature RNA fragment, which is very low. The authors mention that these results are "Consistent with previous observations indicating that more than 80% of the exL form is degraded in vivo" citing the article Tseng et al., Cell Reports, (2015). It is not clear where this information comes from in the cited article. Furthermore, it is surprising that 80% of the hTR precursors would be wasted for nothing, and would not produce a mature hTR.

Figure 1C: In this figure, the authors use a biotinylated full-length exL hTR RNA with a 5' monoguanosine cap to pull-down hTR-associated proteins. However, it is not clear if its kinetic of 3' end processing and accumulation of mature hTR RNA is identical to the processing of the shorter exL RNA used in Figure 1A. Also, according to the data of Figure 1B, 80% of the exL hTR precursor is degraded and only 20% is converted to the mature hTR. If only 20% of the RNA is left after 120 minutes, why the amount of DHX36, DKC1 and TCAB1 proteins recovered from the pull-down remains constant over time? I would expect that less proteins would be recovered as the RNA get degraded?

Figure 2C: The authors should check by RT-qPCR if hTR is associated or not with the LARP proteins after immunoprecipitation.

Figure 4B: The authors do not show if there's an impact of LARP3 depletion on the accumulation of the extended form of hTR in vivo. There's no comparison with a control shRNA, like in Figure 5B.

Figure 5: The authors do not quantify the expression level of the LARP3 protein in K562 cells versus 293T cells. Is there a clear difference of expression between the two cell lines?

Figure 5C and D: The total level of mature hTR is only 2% of the extended form (exL hTR) following treatment with extract from K562 cells! One can see a decrease of DHX36 and DKC1 pull-down over time (see Figure 5D). Why is it different from the results in Figure 1C. Is it cell-specific?

Figure 5F: The impact of LARP3 depletion on telomeres length varies significantly depending on the clone tested. Telomere restriction fragment (TRF) length should be measured and reported. Also, is there an effect of the number of cell passages on telomeres length?

Figure 6A: Depletion of MePCE does not look very efficient. This should be quantified.

Figure 6B: There's no control shRNA to assess the impact of LARP7 or MePCE on the expression of total hTR or the exL form.

Figure 6E: A telomerase activity assay is the key missing experiment here. What is the impact of the depletion of LARP7 or MePCE on telomerase activity? Also, the impact of LARP7 or MePCE depletion on telomeres length is quite variable depending on the clones obtained (see Supplementary Figure 5C). Quantification of telomere restriction fragment length would help answer this question.

Figure 6F: There's a concern regarding the RNA FISH used to detect hTR RNA. There are very few foci visible in cells depleted of PARN, LARP7 or MePCE, and the ones in the cytoplasm, pointed by arrows, look more like "clumps". More convincing data should be provided here.

Reviewer #2 (Remarks to the Author):

In their manuscript titled "LARP3, LARP7, and MePCE are Involved in the Early Stage of Human Telomerase RNA Biogenesis" the authors analyze the contribution of the La-related proteins (LARPs) LARP3, LARP7, and MePCE and the maturation of the human telomerase RNA (hTR) 3'-end. The authors develop a new in vitro telomerase assembly and maturation assay in which in vitro transcribed hTR variants are incubated with cell lysates and analyzed by length changes or used for RNA pull-downs to analyze the association of various telomerase ribonucleoprotein factors. The authors are able to recapitulate the successive binding of telomerase biogenesis co-factors and directly visualize successive exonucleolytic processing of hTR. For instance, NAF1 binds to hTR before GAR1, and then dissociates, while GAR1 continues to accumulate, and an extended long form exL, is first processed into the extended short (exS) form before forming the mature hTR. Using this telomerase assembly assay, the authors go on to show that LARP3, LARP7, and MePCE associated with hTR in the initial stages of maturation and do not bind to mature, catalytically active telomerase. Importantly, this binding depends on the presence of 3' extended forms of hTR. Next, the authors demonstrate that LARP3 binding and triple helix formation by the exL form of TR compete with each other, and LARP3 overexpression counteracts hTR maturation and reduces telomerase activity. In contrast, LARP3 depletion increases hTR levels and telomerase activity, together suggesting that LARP3 binding inhibits telomerase biogenesis. These findings are further supported by an increase in telomere length in LARP3 depleted cells. Depletion of LARP7 and MePCE, attenuated hTR maturation, decreased telomere length, and lead to an apparent accumulation of hTR in cytoplasmic foci. Altogether, the data presented is of very high quality, the findings presented significantly expand our understanding of telomerase assembly, and for the first time describe a method to analyze telomerase assembly in human cell extracts. Throughout the manuscript the authors make many quantitative comparisons between various experimental conditions that lack proper quantification and statistical analyses for significance. In addition, key method details are currently missing. Once those issues are addressed, I fully support publication of this manuscript.

Major Points

1. All claims of differences in the rates of assembly, levels of telomerase activity, and the order of telomerase factor binding need to be supported by statistical analyses. For instance, the authors state that telomerase in the presence of exL hTR is less active than telomerase assembled with mature TR. To make this claim the activity assays have to be carried out in triplicate and normalized to the level of

TR present. Processivity does not have to be normalized but still requires repeats and statistical analysis. The authors also state LARP3 binds to hTR before LARP7 and MePCE, which cannot be concluded from the data presented without careful quantification. LARP3, LARP7, and MePCE all increase from the 0 to the 10-minute time point, and only of the relative increase is less for LARP3 could this conclusion be drawn. The same concern is true for the fraction of cells with cytoplasmic TR foci. I suggest the authors support all of their quantitative comparisons with the appropriate statistical tests.

2. A key achievement of the authors is the development of a new assay that allows assembly of telomerase with human cell extracts. A detailed description of the method to generate the whole cell extracts should be included. Currently no information is included how these extracts were generated.
3. The authors suggest that triple helix formation and binding of LARP3 to the UUU stretch in exL hTR compete. U460C stabilizes the triple helix and disrupts the UUU stretch that LARP3 has been shown to bind to in other RNAs. The authors could make a mutation of the UUU stretch that does not lead to triple helix stabilization to dissect the relative contribution of these two effects.
4. There are no method details on how telomerase activity and processivity were calculated.

Minor Points

1. All abbreviations need to be defined when first introduced.
2. Unclear sentence "Bmc1, a human MePCE homologue, is a fission yeast telomerase holoenzyme."
3. Unclear sentence "The binding of DHX36 to hTR exerted a minor effect on the maturation of the hTR 3' end (Fig. 1c, lanes 8-12)."
4. Page 10 line 3 needs to reference Fig 3c not just the lane number.

Reviewer #3 (Remarks to the Author):

The proper maturation of the human telomerase RNA to form functional telomerase complexes is required for genome integrity, and of clinical importance. This work from Kao et al investigates the role of LARP3, LARP7, and MePCE in telomerase RNA (hTR) biogenesis. They begin by further characterizing a cell lysate system from their prior work and show that incubation of in vitro transcribed hTR in cell lysates leads to a stepwise progression through hTR 3' end processing states, correlating with the association of known H/ACA factors. Here they use this assay to interrogate candidate hTR interaction partners LARP3, LARP7 and MePCE in early stages of hTR maturation. They show effects from manipulating LARP3 on the maturation of hTR in vitro. They also show that shRNA-knockdown of LARP3, LARP7, and MePCE can lead to changes in telomere length in cells. Collectively, the authors use these in vitro and cell-based methods to investigate human telomerase RNA biogenesis, and propose roles for LARP3, LARP7 and MePCE.

The manuscript is interesting and the quality of the data provided is high. However in several cases the interpretations and conclusions the authors draw about the roles of LARP3, LARP7 and MePCE in hTR biogenesis and in telomere length control do not appear to be fully supported by the data presented.

Main comments:

1. A negative regulatory role in early hTR maturation is proposed for LARP3, based on its association with exL and faster kinetics of hTR maturation with LARP3 knockdown (Figure 3). The mechanism hypothesized is LARP3 association with a UUU motif required for tertiary structure formation. The authors test this using a U460C mutation of this motif and the effects of the U460C mutation on LARP3 binding and hTR maturation are clear. However, these experiments do not show that LARP3 alters the kinetics or frequency of hTR tertiary structure formation (pg 10 lines 4-7, title of figure). Thus the claim that LARP3 competes with tertiary structure formation is not well supported. The effect seen could be entirely from binding of UUU, a known aspect of LARP3, and one that may impact the in vitro system irrespective of a role in cells. What effect do changes in hTR triple helix structure

propensity without changes in this motif (as shown in the Tseng, et al 2018) paper have on LARP3 association and maturation? Can biophysical evidence of LARP3 effects on the triple helix structure in vitro be demonstrated?

2. LARP3 shRNA appears to have different effects on hTR steady state levels in 293T and K562 cells (Figures 4 and 5). In 293T cells, LARP3 knockdown had no effects and overexpression increased hTR levels, while in K562 cells, LARP3 shRNA increased hTR levels. Further, in 293T cells, knockdown did not change the amount of dyskerin recovered in vitro in pulldowns, while in K562, this was increased. Replication of loss-of-function in cells and the inconsistencies should be addressed more rigorously prior to drawing conclusions about effects of LARP3 on telomerase and telomeres.

- A. How can the robust effect of manipulating LARP3 on hTR end processing in vitro and in 293T cells (3x higher levels of the exL with LARP3 overexpression) be reconciled with the minimal effect on steady state level of hTR in 293T cells?
- B. Quantitation of total hTR using Northern blot rather than qPCR should be performed
- C. Inconsistent results between cell lines brings up the possibility of off-target effects. Additional shRNAs and/or CRISPR/Cas9 gene inactivation should be used.
- D. Telomere length changes in more than one cell line (e.g. 293T) along with timelines and kinetics in the methods, and the effects of LARP3 overexpression over time, could also strengthen support of the impact being via changes in hTR.

3. Similarities are drawn between the effects of LARP7 and MePCE knockdown on hTR end processing and cytoplasmic localization with the effects observed in PARN deficient cells (Figure 6). However it is well known that in PARN deficient cells, hTR steady state levels are reduced, whereas shRNAs targeting LARP7 or MePCE did not reduce hTR steady state levels. Moreover the cytosolic localization should destabilize hTR. These observations are not consistent with the authors' model.

- A. The authors suggest that inhibition of LARP7 and MePCE leads to a PARN-deficient like phenotype in the end processing assay, but this effect looks subtle in the data presented (Figure 6d). Can this effect be seen / compared to PARN-deficiency using more sensitive methods such as sequencing or RACE as in their prior work?
- B. As for LARP3, the strength of the studies and conclusions drawn in cell based assays on telomeres would benefit from more than single shRNAs and/or other loss-of-function approaches

Overall, the manuscript supports a potential new role for LARP3/LARP7/MePCE in hTR biogenesis but with inconsistent results of manipulating LARP3/LARP7/MePCE on hTR levels and telomeres in cells that requires further elaboration and investigation.

Additional comments:

1. In Figure 1, it is surprising that the abundance of DXH36 is nearly constant in the IP'd samples during the reactions, even though the total amount of RNA recovered is reduced by ~80% between 0 minutes and 120 minutes (Figure 1B). A northern blot probed for hTR / with streptavidin would be helpful to show the results of these experiments (truncated/full length versions) are in agreement, and would aid in the interpretation of these western blots.

2. In Figure 1d, the inclusion of loading control which quantifies the amount of telomerase complexes present in the reaction (such as dyskerin WB) is critical for dissociating differences in the abundance of telomerase vs the function of the recovered telomerase (eg, claim of increased processivity on pg 7 lines 18-19).

3. In Figure 2, additional experimental details and controls are needed.

- A. hTR quantification in 2a as a recovery control would aid in interpretation of conclusions about changes in protein abundance (pg. 8 lines 7-8).
- B. Incubation time in figure 2b should be provided.
- C. For figure 2c, western blotting showing the efficiency of IP would be helpful for interpretation

4. In the Discussion the authors state that short telomere length and high LARP3 expression are seen in poor-prognosis CML, and thus the authors propose that LARP3 inhibition and telomere lengthening could be therapeutically useful. However, these correlations do not imply that reversing LARP3 expression or especially increasing telomere length would alter the pathological state of the cell (the latter could risk extending replicative capacity of cancer cells). CML in chronic phase is an indolent process that evolves to blast crisis due to multiple genetic hits, which more likely underlie the changes in gene expression and telomere attrition in these rapidly dividing cells, rather than short telomeres being the driver of blast crisis. This paragraph should be revised or deleted.

REVIEWER COMMENTS

Reviewer #1 (Remarks to the Author):

In this manuscript, Kao and colleagues report that the La-related proteins LARP3, LARP7 and MePCE are involved in the early processing of the telomerase RNA hTR during telomerase biogenesis in human cells. Previous studies revealed that La-related proteins are involved in the 3' end processing of telomerase RNA in the fission yeast and ciliates. So far, no evidence for a role of LARP proteins in the biogenesis of human telomerase has been reported yet. This manuscript now shows that the human La-related proteins LARP3, LARP7 and MePCE are also involved in the regulation of hTR 3' end processing, control the assembly of telomerase holoenzyme and impact telomeres length homeostasis. However, this manuscript falls short of providing convincing experimental evidence to support the claims made by the authors. Several controls are missing and some results are contradictory (see major comments below). As it is, this manuscript is not acceptable for publication in this journal, and would require major modifications in order to be publishable.

We thank the reviewer for complementing our work and the thoughtful suggestions on how to improve specific aspects. We have modified the manuscript accordingly and addressed the individual points below.

Major comments

Figure 1A: To what extent this assay faithfully reproduces the 3' end processing of hTR *in vivo*? The authors do not provide evidence that their *in vitro* hTR maturation assay depends on the exosome and the exonucleases PARN and TOE1. The authors should validate their assay by depleting exosome components, PARN and TOE1 using RNAi. Furthermore, it is not clear that exL RNA processing in this *in vitro* assay occurs exclusively from the 3' end. Is it possible that some of the degradation occurs from the 5' end?

As shown in our previous publication (Tseng et al., 2018), *in vitro* 3' end processing assays showed that oligo-adenylated 3'-extended hTR was deadenylated but not processed into mature hTR following knockdown of PARN or RRP6, suggesting that the *in vitro* hTR maturation assay depends on the 3' end processing factors, at least RRP6 and PARN (Figure 2c and 3b in Tseng et al., 2018). To make it clear, we have added statement "The requirement of RRP6 and PARN for hTR 3' end maturation has been supported by the *in vitro* cell-free system. Knockdown of RRP6 and PARN impaired the 3' end processing of hTR." in the introduction section.

We agree with the reviewer 1 that some of the degradation occurs from the 5' end in our *in vitro* assay. As shown in new supplemental Figure 1C, in this regard, the full length of biotinylated exL with a monoguanosine cap was pulled down. when the probe matching the mature 3' end of hTR, the truncated version of hTR was observed. The 5' to 3' degradation pathway has been suggested *in vivo* through DCP2 and XRN1 (Shukla et al., 2016).

Reference paper:

1. Tseng, C.K., Wang, H.F., Schroeder, M.R., and Baumann, P. (2018). The H/ACA complex disrupts triplex in hTR precursor to permit processing by RRP6 and PARN. *Nat Commun* 9, 5430. 10.1038/s41467-018-07822-6.
2. Shukla, S., Schmidt, J.C., Goldfarb, K.C., Cech, T.R., and Parker, R. (2016). Inhibition of telomerase RNA decay rescues telomerase deficiency caused by dyskerin or PARN defects. *Nat Struct Mol Biol* 23, 286-292. 10.1038/nsmb.3184.

Figure 1B: The majority (80%) of exL RNA is degraded in the *in vitro* assay, and only 20% is converted to the mature RNA fragment, which is very low. The authors mention that these results are "Consistent with previous observations indicating that more than 80% of the exL form is degraded *in vivo*" citing the article Tseng et al., *Cell Reports*, (2015). It is not clear where this information comes from in the

cited article. Furthermore, it is surprising that 80% of the hTR precursors would be wasted for nothing, and would not produce a mature hTR.

As shown in Figure 2d in Cell Rep, 2015, when the additional H/ACA box was placed 50 nt downstream of position 451, 3'-extended forms of ~750 nt were formed without the expense of the 451-nt mature form, suggesting that 3' extended hTR is predominantly degraded. Currently why the majority of the longer transcripts are not converted into the mature form remains to be investigated. To make it clear, we have changed the statement to read "Previous studies suggest that long transcripts are predominantly degraded in vivo. Supporting these observations, more than 80% of the exL form was degraded and only 20% of the exL form was converted into the mature form of hTR in the *in vitro* assay".

Figure 1C: In this figure, the authors use a biotinylated full-length exL hTR RNA with a 5' monoguanosine cap to pull-down hTR-associated proteins. However, it is not clear if its kinetic of 3' end processing and accumulation of mature hTR RNA is identical to the processing of the shorter exL RNA used in Figure 1A. Also, according to the data of Figure 1B, 80% of the exL hTR precursor is degraded and only 20% is converted to the mature hTR. If only 20% of the RNA is left after 120 minutes, why the amount of DHX36, DKC1 and TCAB1 proteins recovered from the pull-down remains constant over time? I would expect that less proteins would be recovered as the RNA gets degraded?

1. The same experiment was performed except that the biotinylated FL exL hTR with a 5' monoguanosine cap was used, followed by the streptavidin pulldown. As shown in the new supplemental figure 1c, after the incubation, total hTR was decreased by more than 80% in 293T cells. These data suggest a similar kinetic between the full-length and shorter forms of hTR.
2. In terms of amounts of proteins recovered from the pull-down, all pull-down experiments were now presented in a statistical manner in all new figures. We normalized individual proteins to the corresponding peak signal. As shown in Figure 1C, the binding of H/ACA complex components (DKC1, NOP10, NHP2, GAR1, and TCAB1) was decreased although with variable efficiency. The difference between RNA degradation and protein recovery could be due to several factors, such as the sensitivity of Northern probes vs antibodies and different binding affinity of protein to RNA.
3. The binding of DHX36 remains nearly constant over time. Given that DHX36 binds to the 5' G-rich region of hTR. These 3'-truncated versions may also contribute to the binding of DHX36.

Figure 2C: The authors should check by RT-qPCR if hTR is associated or not with the LARP proteins after immunoprecipitation.

As shown in the new Figure 2d, only small amounts of hTR could be co-precipitated with LARP3, LARP7, and MePCE compared to DKC1. These data support the observations that low telomerase activity of LARP3-, LARP7, and MePCE-associated telomerases were detected as shown in Figure 2e.

Figure 4B: The authors do not show if there's an impact of LARP3 depletion on the accumulation of the extended form of hTR in vivo. There's no comparison with a control shRNA, like in Figure 5B.

In old figure 4B, the fold change has been relative to their control. We skipped the control bar. To reduce confusion, we put the control bar back on the bar graph as shown in the new Figure 4b and 4c.

Figure 5: The authors do not quantify the expression level of the LARP3 protein in K562 cells versus 293T cells. Is there a clear difference of expression between the two cell lines?

We checked the expression level of the LARP3 protein in K562 and 293T cells. There is no significant difference between these two cell lines. It is always difficult to compare two different cell lines. However, the evaluated expression of LARP3 has been reported in various cancers, including CML (Sommer and Heise, 2021).

Reference paper:

Sommer, G., and Heise, T. (2021). Role of the RNA-binding protein La in cancer pathobiology. *RNA Biol.* 18, 218–236. [10.1080/15476286.2020.1792677](https://doi.org/10.1080/15476286.2020.1792677).

Figure 5C and D: The total level of mature hTR is only 2% of the extended form (exL hTR) following treatment with extract from K562 cells! One can see a decrease of DHX36 and DKC1 pull-down over time (see Figure 5D). Why is it different from the results in Figure 1C. Is it cell-specific?

The degradation rate of hTR in K562 cells (new Fig. 5c) is much faster than that in 293T cells (new Fig. 1b). The difference in degradation rate was also observed in the full length of biotinylated exL with a monoguanosine cap (new supplementary 1c). The difference in hTR stability of these two cell lines could be cell-specific. The obviously decrease in the amount of DHX36 could result from the fast degradation efficiency of hTR in K562 cells.

Figure 5F: The impact of LARP3 depletion on telomeres length varies significantly depending on the clone tested. Telomere restriction fragment (TRF) length should be measured and reported. Also, is there an effect of the number of cell passages on telomeres length?

The effect of the number of cell passages and different clones on telomere length is now showed in the new Figure 5F.

Figure 6A: Depletion of MePCE does not look very efficient. This should be quantified.

We consider poor knockdown efficiency of MePCE, we have repeated PARN, LARP7, MePCE knockdown in HeLa cells and quantified as shown in new figure 6a. The results show at least 90% reductions at protein levels of PARN, LARP7, and MePCE.

Figure 6B: There's no control shRNA to assess the impact of LARP7 or MePCE on the expression of total hTR or the exL form.

To reduce confusion and make our results more convincing, we measured the expression of hTR and 3' end distribution of hTR in our new knockdown cells. For hTR expression, we put the control bar back on the bar graph as shown in the new Figure 6e. The new Figure 6f shows the 3' end distribution. The results shows that knockdown of LARP7 and MePCE resulted in an increase in the fraction of longer hTR transcripts with 3' termini mapping beyond position 451 (new Fig. 6f).

Figure 6E: A telomerase activity assay is the key missing experiment here. What is the impact of the depletion of LARP7 or MePCE on telomerase activity? Also, the impact of LARP7 or MePCE depletion on telomeres length is quite variable depending on the clones obtained (see Supplementary Figure 5C). Quantification of telomere restriction fragment length would help answer this question.

As shown in the new figure 6c, knockdown of LARP7 and MePCE caused a reduction in telomerase activity. The telomere length of LARP7 or MePCE knockdown HeLa cells (New Figure 6b) and 293T cells (Supplementary Fig. 9b) shortened. Quantification of telomere length has been done as shown in the new figures.

Figure 6F: There's a concern regarding the RNA FISH used to detect hTR RNA. There are very few foci visible in cells depleted of PARN, LARP7 or MePCE, and the ones in the cytoplasm, pointed by arrows, look more like "clumps". More convincing data should be provided here.

We agree with the reviewer. We did cell fractionation and checked the localization of hTR by Northern blot. Consistent with the findings of the imaging experiments, cell fractionation experiments revealed increases in the fractions of cytoplasmic hTR upon PARN, LARP7, and MePCE knockdown (new Fig. 6i).

Reviewer #2 (Remarks to the Author):

In their manuscript titled “LARP3, LARP7, and MePCE are Involved in the Early Stage of Human Telomerase RNA Biogenesis” the authors analyze the contribution of the La-related proteins (LARPs) LARP3, LARP7, and MePCE and the maturation of the human telomerase RNA (hTR) 3'-end. The authors develop a new in vitro telomerase assembly and maturation assay in which in vitro transcribed hTR variants are incubated with cell lysates and analyzed by length changes or used for RNA pull-downs to analyze the association of various telomerase ribonucleoprotein factors. The authors are able to recapitulate the successive binding of telomerase biogenesis co-factors and directly visualize successive exonucleolytic processing of hTR. For instance, NAF1 binds to hTR before GAR1, and then dissociates, while GAR1 continues to accumulate, and an extended long form exL, is first processed into the extended short (exS) form before forming the mature hTR. Using this telomerase assembly assay, the authors go on to show that LARP3, LARP7, and MePCE associated with hTR in the initial stages of maturation and do not bind to mature, catalytically active telomerase. Importantly, this binding depends on the presence of 3' extended forms of hTR. Next, the authors demonstrate that LARP3 binding and triple helix formation by the exL form of TR compete with each other, and LARP3 overexpression counteracts hTR maturation and reduces telomerase activity. In contrast, LARP3 depletion increases hTR levels and telomerase activity, together suggesting that LARP3 binding inhibits telomerase biogenesis. These findings are further supported by an increase in telomere length in LARP3 depleted cells. Depletion of LARP7 and MePCE, attenuated hTR maturation, decreased telomere length, and lead to an apparent accumulation of hTR in cytoplasmic foci. Altogether, the data presented is of very high quality, the findings presented significantly expand our understanding of telomerase assembly, and for the first time describe a method to analyze telomerase assembly in human cell extracts. Throughout the manuscript the authors make many quantitative comparisons between various experimental conditions that lack proper quantification and statistical analyses for significance. In addition, key method details are currently missing. Once those issues are addressed, I fully support publication of this manuscript.

We would like to thank reviewer 2 for a strongly positive assessment of our work and for stating that this study “the data presented is of very high quality, the findings presented significantly expand our understanding of telomerase assembly, and for the first time describe a method to analyze telomerase assembly in human cell extracts”.

Major Points

1. All claims of differences in the rates of assembly, levels of telomerase activity, and the order of telomerase factor binding need to be supported by statistical analyses. For instance, the authors state that telomerase in the presence of exL hTR is less active than telomerase assembled with mature TR. To make this claim the activity assays have to be carried out in triplicate and normalized to the level of TR present. Processivity does not have to be normalized but still requires repeats and statistical analysis.

We have now done all statistical analyses shown in all new figure 1f. We have now included two additional biological replicates as new supplementary Figure 2b and 2c.

The authors also state LARP3 binds to hTR before LARP7 and MePCE, which cannot be concluded from the data presented without careful quantification. LARP3, LARP7, and MePCE all increase from the 0 to the 10-minute time point, and only of the relative increase is less for LARP3 could this conclusion be drawn.

We did quantification as shown in the new Figure 2a. We agree with the reviewer 2. We changed the statement to read “LARP3 associated with hTR concurrently with binding of LARP7 and MePCE (Fig. 2a, lanes 8-10).”

The same concern is true for the fraction of cells with cytoplasmic TR foci. I suggest the authors support all of their quantitative comparisons with the appropriate statistical tests.

We agree with the reviewer. We did quantitative comparisons with the appropriate statistical tests (new fig. 6h). To provide more convincing data, we did cell fractionation and checked the localization of hTR by Northern blot. Consistent with the findings of the imaging experiments, cell fractionation experiments revealed increases in the fractions of cytoplasmic hTR upon PARN, LARP7, and MePCE knockdown (new Fig. 6i).

2. A key achievement of the authors is the development of a new assay that allows assembly of telomerase with human cell extracts. A detailed description of the method to generate the whole cell extracts should be included. Currently no information is included how these extracts were generated.

We have included this information in our new materials and methods. As shown in “Cell extracts preparation” and also shown below:

Human cells extracts were prepared from cell pellets (2×10^7 cells) with 250 μ l CHAPS buffer (0.5% CHAPS; 50 mM Tris-HCl, pH 8.0; 50 mM KCl; 1 mM $MgCl_2$; 1 mM EGTA; 10% glycerol; 5 mM DTT; 1 mM PMSF). The cells be lysed at 4°C on a nutator for 1 hour. The lysate was then centrifuged at 4°C at 15,000 r.p.m. for 10 minutes. The supernatant was transferred to a new tube and centrifuged at 4°C at 15,000 r.p.m. for another 10 minutes. The protein concentrations were measured by Bradford Assay (Bio-Rad). Cell extracts were stored at -80°C.

3. The authors suggest that triple helix formation and binding of LARP3 to the UUU stretch in exL hTR compete. U460C stabilizes the triple helix and disrupts the UUU stretch that LARP3 has been shown to bind to in other RNAs. The authors could make a mutation of the UUU stretch that does not lead to triple helix stabilization to dissect the relative contribution of these two effects.

To see whether the amount of LARP3 could be pulled down with hTR that retains the UUU stretch but could not form triple helix, we disrupted the tertiary base interactions but retained the 3'-terminal UUU sequence by changing the box H sequence (372-AGAGGA-377) to the disease-related box H mutant (375-377GGAdel) hTR that disrupts the interaction between the 3'-terminal UUU and box H (New figure 3a). This mutant was identified in patients with idiopathic pulmonary fibrosis (Alder, J.K. et al., 2011). As shown in the new Figure 3, the 375-377GGAdel mutant, which was expected to have an exposed terminal U sequence and disrupted triple-helix formation, pulled down more LARP3 (new Fig. 3b, lane 14), and LARP3 continued to bind to the 375-377GGAdel mutant relatively longer than to the wild-type hTR (new Fig. 3c, lanes 18-22), which blocked the conversion of exL to its mature form and resulted in its degradation (new Fig. 3d, lanes 11-20 and new Supplementary Fig. 4b). These data suggest that LARP3 binding to the UUU stretch of exL hTR competes with the formation of tertiary structures in the exL form, determining the efficiency of mature hTR production.

Reference paper:

Alder, J.K. et al. Telomere length is a determinant of emphysema susceptibility. *Am J Respir Crit Care Med* 184, 904-12 (2011).

4. There are no method details on how telomerase activity and processivity were calculated.

We have added the details in the “Materials and Methods” and also shown below:

For telomerase activity measurement, the intensity of all bands was measured and normalized to that of the control. For telomerase processivity measurements, the intensity of each major repeat band was measured and normalized to both the intensity of the first band and the number of ^{32}P -labelled nucleotides incorporated. Normalized intensities were then plotted versus the number of repeats.

Minor Points

1. All abbreviations need to be defined when first introduced.

Done

2. Unclear sentence “Bmc1, a human MePCE homologue, is a fission yeast telomerase holoenzyme.”

To make it clear, we have changed the statement to read "Bin3/MePCE 1 (Bmc1) is the putative fission yeast orthologue of MePCE and cooperates with Pof8 to recognize correctly folded TER1"

3. Unclear sentence "The binding of DHX36 to hTR exerted a minor effect on the maturation of the hTR 3' end (Fig. 1c, lanes 8-12)."

To make it clear, we have changed the statement to read "The binding of DHX36 to hTR remained constant during the process of 3' end maturation"

4. Page 10 line 3 needs to reference Fig 3c not just the lane number.

We have referenced.

Reviewer #3 (Remarks to the Author):

The proper maturation of the human telomerase RNA to form functional telomerase complexes is required for genome integrity, and of clinical importance. This work from Kao et al investigates the role of LARP3, LARP7, and MePCE in telomerase RNA (hTR) biogenesis. They begin by further characterizing a cell lysate system from their prior work and show that incubation of in vitro transcribed hTR in cell lysates leads to a stepwise progression through hTR 3' end processing states, correlating with the association of known H/ACA factors. Here they use this assay to interrogate candidate hTR interaction partners LARP3, LARP7 and MePCE in early stages of hTR maturation. They show effects from manipulating LARP3 on the maturation of hTR in vitro. They also show that shRNA-knockdown of LARP3, LARP7, and MePCE can lead to changes in telomere length in cells. Collectively, the authors use these in vitro and cell-based methods to investigate human telomerase RNA biogenesis, and propose roles for LARP3, LARP7 and MePCE.

The manuscript is interesting and the quality of the data provided is high. However in several cases the interpretations and conclusions the authors draw about the roles of LARP3, LARP7 and MePCE in hTR biogenesis and in telomere length control do not appear to be fully supported by the data presented.

We thank the reviewer for complementing our work and the thoughtful suggestions on how to improve specific aspects. We have modified the manuscript accordingly and addressed the individual points below.

Main comments:

1. A negative regulatory role in early hTR maturation is proposed for LARP3, based on its association with exL and faster kinetics of hTR maturation with LARP3 knockdown (Figure 3). The mechanism hypothesized is LARP3 association with a UUU motif required for tertiary structure formation. The authors test this using a U460C mutation of this motif and the effects of the U460C mutation on LARP3 binding and hTR maturation are clear. However, these experiments do not show that LARP3 alters the kinetics or frequency of hTR tertiary structure formation (pg 10 lines 4-7, title of figure). Thus the claim that LARP3 competes with tertiary structure formation is not well supported. The effect seen could be entirely from binding of UUU, a known aspect of LARP3, and one that may impact the in vitro system irrespective of a role in cells. What effect do changes in hTR triple helix structure propensity without changes in this motif (as shown in the Tseng, et al 2018) paper have on LARP3 association and maturation? Can biophysical evidence of LARP3 effects on the triple helix structure in vitro be demonstrated?

We agree with the reviewer that the binding of LARP3 to exL could depend on the terminal U stretch according to our new results. The conclusion also suggested that the binding of LARP3 to the terminal U stretch competes with tertiary structure formation of exL. Please see below.

To see whether the amount of LARP3 could be pulled down with hTR that retains the UUU stretch but could not form triple helix, we disrupted the tertiary base interactions but retained the 3'-terminal UUU sequence by changing the box H sequence (372-AGAGGA-377) to the disease-related box H mutant (375-377GGAdel) hTR that disrupts the interaction between the 3'-terminal UUU and box H (New figure 3a). This mutant was identified in patients with idiopathic pulmonary fibrosis (Alder, J.K. et al., 2011). As shown in the new Figure 3, the 375-377GGAdel mutant, which was expected to have an exposed terminal U sequence and disrupted triple-helix formation, pulled down more LARP3 (new Fig. 3b, lane 14), and LARP3 continued to bind to the 375-377GGAdel mutant relatively longer than to the wild-type hTR (new Fig. 3c, lanes 18-22), which blocked the conversion of exL to its mature form and resulted in its degradation (new Fig. 3d, lanes 11-20 and new Supplementary Fig. 4b). These data suggest that LARP3 binding to the UUU stretch of exL hTR competes with the formation of tertiary structures in the exL form, determining the efficiency of mature hTR production.

Reference paper:

Alder, J.K. et al. Telomere length is a determinant of emphysema susceptibility. *Am J Respir Crit Care Med* 184, 904-12 (2011).

2. LARP3 shRNA appears to have different effects on hTR steady state levels in 293T and K562 cells (Figures 4 and 5). In 293T cells, LARP3 knockdown had no effects and overexpression increased hTR levels, while in K562 cells, LARP3 shRNA increased hTR levels. Further, in 293T cells, knockdown did not change the amount of dyskerin recovered in vitro in pulldowns, while in K562, this was increased. Replication of loss-of-function in cells and the inconsistencies should be addressed more rigorously prior to drawing conclusions about effects of LARP3 on telomerase and telomeres.

- A. How can the robust effect of manipulating LARP3 on hTR end processing in vitro and in 293T cells (3x higher levels of the exL with LARP3 overexpression) be reconciled with the minimal effect on steady state level of hTR in 293T cells?

The steady state level of hTR detected is primarily the ~451 nt RNA species. The amount of endogenous exL in cells is much lower. We only "transiently" overexpressed LARP3 in 293T cells. Therefore, even a dramatic increase in the small fractions of exL may not affect the steady state level of hTR. Whether a long-term expression of LARP3 could cause obvious effects remains to be investigated.

- B. Quantitation of total hTR using Northern blot rather than qPCR should be performed

We guess the reviewer mentions RT-qPCR, not qPCR. According to many literatures. Amount of RNA detected by Northern blot and RT-qPCR would give the same results. On one hand, we think RT-qPCR would be a better way than Northern blot for RNA quantitation. On the other hand, Northern blot mainly present ~451 nt products but could not easily detect RNA with heterogeneous ends. Therefore, we use RT-qPCR instead of Northern blot for total RNA quantitation.

- C. Inconsistent results between cell lines brings up the possibility of off-target effects. Additional shRNAs and/or CRISPR/Cas9 gene inactivation should be used.

As shown in the new Supplementary Figure 5, three different shRNAs targeting LARP3 give the same results although the efficiencies are different.

- D. Telomere length changes in more than one cell line (e.g. 293T) along with timelines and kinetics in the methods, and the effects of LARP3 overexpression over time, could also strengthen support of the impact being via changes in hTR.

Wright Lab has previously demonstrated that overexpression of LARP3 in experimentally immortalized human cells and prostate cancer cells results in gradual telomere shortening (Ford, L.P., et. al. 2001). Our data here indicate that overexpression of LARP3 impairs hTR biogenesis. These data suggest that high expression levels of LARP3 have negative effect on telomere homeostasis and reducing expression levels of LARP3 in cell line with evaluated levels of LARP3, such as K562 cells, could restore telomere length. As shown in the new Figure 5f, we did see telomeres elongation in K562 along with kinetics. However, telomere did not elongate in 293T cells upon LARP3 knockdown (the new Supplementary Figure 6). To our knowledge, no report suggests that the evaluated levels of LARP3 was observed in 293T cells. Therefore, these data suggest to us that no negative effect of LARP3 on hTR biogenesis in

293T cells and also suggest that LARP3 effects on telomere homeostasis is cell-specific. Whether these effects could be observed in other cancer types that have high levels of LARP3 remains to be investigated in the future.

Reference:

Ford, L.P., Shay, J.W. & Wright, W.E. The La antigen associates with the human telomerase ribonucleoprotein and influences telomere length in vivo. *RNA* 7, 1068–75 (2001).

3. Similarities are drawn between the effects of LARP7 and MePCE knockdown on hTR end processing and cytoplasmic localization with the effects observed in PARN deficient cells (Figure 6). However it is well known that in PARN deficient cells, hTR steady state levels are reduced, whereas shRNAs targeting LARP7 or MePCE did not reduce hTR steady state levels. Moreover the cytosolic localization should destabilize hTR. These observations are not consistent with the authors' model.

We agree with the reviewer. We consider poor knockdown efficiency of LARP7 and MePCE, we have repeated PARN, LARP7, MePCE knockdown in HeLa cells and quantified as shown in new figure 6a and new Supplementary Figure 8. The results show at least 90% reductions at protein levels of PARN, LARP7, and MePCE. Now we do see reductions in the steady-state levels of hTR (new figure 6e and new Supplementary Figure 8).

- A. The authors suggest that inhibition of LARP7 and MePCE leads to a PARN-deficient like phenotype in the end processing assay, but this effect looks subtle in the data presented (Figure 6d). Can this effect be seen / compared to PARN-deficiency using more sensitive methods such as sequencing or RACE as in their prior work?

We agree with the reviewer. We have analyzed the 3' end distribution of hTR upon PARN, LARP7, and MePCE. As shown in new figure 6f, PARN, LARP7, and MePCE knockdown causes accumulation of 3'-extended hTR compared to the shLuc control. These data support our in vitro 3' processing assay (new supplementary 9d)

- B. As for LARP3, the strength of the studies and conclusions drawn in cell based assays on telomeres would benefit from more than single shRNAs and/or other loss-of-function approaches. As shown in the new Supplementary Figure 5, three different shRNAs targeting LARP3 give the same results although the efficiencies are different. LARP3 knockdown booster the hTR biogenesis and telomerase activity. These phenotypes were seen in 293T cells (this study), K562 cells (this study), and iPSC (data not shown, it will be our follow-up paper), suggesting a negative role for LARP3 in telomerase biogenesis.

Overall, the manuscript supports a potential new role for LARP3/LARP7/MePCE in hTR biogenesis but with inconsistent results of manipulating LARP3/LARP7/MePCE on hTR levels and telomeres in cells that requires further elaboration and investigation.

Additional comments:

1. In Figure 1, it is surprising that the abundance of DXH36 is nearly constant in the IP'd samples during the reactions, even though the total amount of RNA recovered is reduced by ~80% between 0 minutes and 120 minutes (Figure 1B). A northern blot probed for hTR / with streptavidin would be helpful to show the results of these experiments (truncated/full length versions) are in agreement, and would aid in the interpretation of these western blots.

As shown in the new supplemental figure 1c, Northern blot using probes matching the hTR 3' end of purified telomerase showed that the kinetic of 3' end processing of capped exL RNA is similar to that of the shorter exL RNA used in Figure 1A. The truncated versions smaller than ~400 nt are nearly constant. Given that DXH36 binds to the 5' G-rich region of hTR. These truncated versions would also contribute to the binding of DXH36.

2. In Figure 1d, the inclusion of loading control which quantifies the amount of telomerase complexes present in the reaction (such as dyskerin WB) is critical for dissociating differences in the abundance

of telomerase vs the function of the recovered telomerase (eg, claim of increased processivity on pg 7 lines 18-19).

We have done and shown in the new figure 1d. In this experiment, we compared the activity between telomerases containing exL or mature forms. The similar amount of DKC1 and hTR was recovered.

3. In Figure 2, additional experimental details and controls are needed.
- A.hTR quantification in 2a as a recovery control would aid in interpretation of conclusions about changes in protein abundance (pg. 8 lines 7-8).
As shown in the new supplementary Figure 1c, the hTR was degraded over time. These data suggest that telomerase components decreased during the incubation results from the turnover of telomerase RNPs. It is difficult to compare protein signals from western blot and heterogeneous hTR signals from northern blot. On the other hand, small fragments of hTR may not be probed due to the hybridization temperature (65°C).

- B.Incubation time in figure 2b should be provided.

We have included this information in the materials and methods. The reaction was incubated for 10 min.

- C.For figure 2c, western blotting showing the efficiency of IP would be helpful for interpretation

We have done and shown in the new figure 2c. In addition, we also examined the hTR levels co-precipitated with the corresponding protein as shown in the new figure 2d.

4. In the Discussion the authors state that short telomere length and high LARP3 expression are seen in poor-prognosis CML, and thus the authors propose that LARP3 inhibition and telomere lengthening could be therapeutically useful. However, these correlations do not imply that reversing LARP3 expression or especially increasing telomere length would alter the pathological state of the cell (the latter could risk extending replicative capacity of cancer cells). CML in chronic phase is an indolent process that evolves to blast crisis due to multiple genetic hits, which more likely underlie the changes in gene expression and telomere attrition in these rapidly dividing cells, rather than short telomeres being the driver of blast crisis. This paragraph should be revised or deleted.

We have deleted.

REVIEWER COMMENTS

Reviewer #1 (Remarks to the Author):

In this revised version of their manuscript, the authors include new experiments to answer major comments from the reviewers and better support their model that LARPs proteins are important players in telomerase biogenesis. They also brought some clarifications in the text to answer the questions raised in the first reviews. I think this manuscript is now acceptable for publication in this journal. Still, I have a few minor corrections to the manuscript:

- page 12, lines 18-19: In the rebuttal, the authors mention that LARP3 levels are similar in HEK293T and K562 cells, so it is not clear why K562 cells correspond to one of the "cells lines expressing high levels of LARP3 leads" as stated here.
- page 16, lines 4-5: Your data do not show that LARP3 binds prior to the binding of LARP7 and MePCE. In the description of the figure 2A, it is mentioned that LARP3 binding occurs concurrently with LARP7 and MePCE. This sentence should be corrected.
- First paragraph, page 15: the results from hTR RNA FISH are overstated, as the difference in nuclear versus cytoplasmic hTR between the shRNA control and cells depleted of PRAN, LARP7 or MePCE is not significant. This should be mentioned in the text.
- Figure 2c: The immunoprecipitations using antibodies against LARP7 or MePCE reveal that other LARPs co-immunoprecipitate strongly with LARP7 and MePCE. This should be discussed in the main text or figure legend.
- Figure 6h: For the sh-LARP7, the sum of the white and black bars is more than 100%. Also, correct the title of Y axis: "The percentage of hTR localizriion"

Reviewer #2 (Remarks to the Author):

In this revised manuscript the authors have largely addressed my concerns and added a significant number of additional experiments. In particular the inclusion of the disease mutant of TR that disrupts the triple helix provides additional insight. There are some remaining concerns below regarding statistical analysis when experimental results are stated to be different, and the manuscript needs to be carefully edited for syntax and clarity.

1. Was the telomerase activity data normalized to the amount of TR present? From the northern blot in figure 1d it appears as if there is a higher level of mature length TR (even though the quantification only indicates a 10% increase). A higher amount of TR could explain the increase in activity. Were the northern blots to analyze TR levels carried out for each biological replicate of the telomerase production to normalize each individual experiment?
2. Figure 3d the difference between the amount of mature TR formed with U460C compared to wildtype needs to be tested for significance or stated to be similar to WT. U460C appears to only slight change the amount of mature TR formed.
3. Telomerase activity in figure 4e needs to be quantified like in figure 1 including replicates and normalizations, if the authors want to state LARP3 KD increased telomerase activity.
4. Figure 5C needs statistical analysis.
5. Telomerase activity in figure 5E needs statistical analysis.
6. Figure 6 D and E need proper Y axis labels stating what is being measured. Fold change of what?
7. Page 13 line 21 I believe should reference 6e instead of 6d.
8. Could figure 6f be presented in a way that groups all reads longer than 451 nt to appreciate the

differences more easily?

9. Figure S9 is discussed quite extensively in the text. The authors should consider moving this data to a main figure.

10. Figure 6h the LARP7 bar graph has a typo. Bottom should read 45% not 55%.

11. Abstract needs to be edited for language, it is hard to follow.

12. Page 7 line 21: It is unclear what "these" is referring to. Please restate the subject of this sentence for clarity.

13. Page 8 lines 20-23 mentions "components" several times referring to the LARP protein family members. This manuscript mentions many telomerase components, referencing back to prior sentences with such language is very confusing, especially for non-expert readers. Unless it is completely unambiguous the names of the specific proteins should be restated in each sentence.

14. Page 10 line 5 "in contrast" could be replaced by "in addition".

15. Figure 4 panels are referenced out of order and appear out of order in the figure. Panel C is below Panel D. Please re-order to make it easier to follow.

16. Page 12 line 16 the sentence starting with "Given" is incomplete, it lacks a main clause.

Reviewer #3 (Remarks to the Author):

Kao and colleagues present a revised manuscript with a number of new experiments and analyses requested by this reviewer and others. Many points are addressed, statistical tests have been added, and some arguments have been strengthened. However there continue to be conclusions not fully supported by data, including some of the new statistical analyses. Overall the manuscript presents an interesting and intricate study on hTR biogenesis, but more careful interpretation is required.

Major points:

1) Regarding the original concern of the conclusion "LARP3 competes with tertiary structure", the authors now provide a boxH deletion that cannot interact with the UUU end and find increased LARP3 association and hTR degradation rather than maturation, leading to re-statement of the conclusion that LARP3 competition with tertiary structures competes with maturation. A simpler alternative explanation that must be considered in this mutant is that boxH disruption leads to lack of DKC1 binding and increased susceptibility to exonucleases. However, in Fig. 3C, loss of the boxH appears to increase DKC1 pulldown in the mature form, going against the literature and structural data on H/ACA RNPs. This warrants explanation.

2) New data and statistical analyses do not fully support statements in the rebuttal and text in a number of cases. Conclusions drawn should reflect this. Examples:

a. Fig 6e: northern blot of LARP7 kd is not significant (ns), but text states there are reproducible reductions in the steady-state level of hTR.

b. Fig 6f: replicates and statistics not provided to support the statement that LARP7 and MeCPE kd resulted in increased fraction of longer transcripts beyond 451 nt

c. Fig 6g,h: subcellular localization quantification is 'ns' across the board but regarded as different in the text

d. Argument is made in the rebuttal and revised text for LARP3 level differences explaining inconsistency of results between 293T and K562 ("CML") cells from LARP3 kd, but then it is stated in the rebuttal that levels were found to be no different

Other points:

1) More description should be provided in the methods or figure legends about the generation and validation of stable shRNA kd and overexpression in cell lines used for telomere length analyses. If these are all done by plasmid transfection and drug selection, for instance, there could be bottle-neck effects of selecting very few cells with integrations that confound bulk telomere length interpretation

2) The sentence structure of new text on page 12 should be corrected

3) In Supp Fig 1c, I believe the right panel truncation label (3') is incorrect

4) In the methods, "Preparation of capped hTR...", it is stated that the reaction contains both bioUTP and 32P-UTP. Please confirm.

REVIEWER COMMENTS

Reviewer #1 (Remarks to the Author):

In this revised version of their manuscript, the authors include new experiments to answer major comments from the reviewers and better support their model that LARPs proteins are important players in telomerase biogenesis. They also brought some clarifications in the text to answer the questions raised in the first reviews. I think this manuscript is now acceptable for publication in this journal. Still, I have a few minor corrections to the manuscript:

We thank the reviewer for complementing our work and the thoughtful suggestions on how to improve specific aspects. We have modified the manuscript accordingly and addressed the individual points below.

- page 12, lines 18-19: In the rebuttal, the authors mention that LARP3 levels are similar in HEK293T and K562 cells, so it is not clear why K562 cells correspond to one of the “cells lines expressing high levels of LARP3 leads” as stated here.

LARP3 expression is correlated with poor clinical prognosis in CML patients and increases during CML progression (chronic phase, the accelerated phase, and blast phase)¹. K562 is the first cell line reported in 1975 from a patient with chronic CML in blast crisis and **is characterized by the presence of the BCR-ABL1 fusion gene**, which serves as a crucial molecular marker in the diagnosis and treatment of CML in clinical practice². A study published by Calabretta lab gave the first hint about the molecular mechanism leading to elevated levels of LARP3 protein in BCR-ABL1-transformed cells². Therefore, K562 cells have been widely used for investigating the BCR/ABL1 signaling and represent the prototypical cell culture model of CML.

Reference paper:

1. Trotta, R. et al. BCR/ABL activates mdm2 mRNA translation via the La antigen. *Cancer Cell* 3, 145–60 (2003).
2. Druker, B. J. et al. Effects of a selective inhibitor of the Abl tyrosine kinase on the growth of Bcr-Abl positive cells. *Nat. Med.* 2, 561–566. (1996)
3. Sommer, G., and Heise, T. (2021). Role of the RNA-binding protein La in cancer pathobiology. *RNA Biol.* 18, 218–236. 10.1080/15476286.2020.1792677.

- page 16, lines 4-5: Your data do not show that LARP3 binds prior to the binding of LARP7 and MePCE. In the description of the figure 2A, it is mentioned that LARP3 binding occurs concurrently with LARP7 and MePCE. This sentence should be corrected.

We agree with the reviewer. We changed the statement to read” LARP3 preferentially bound to the hTR precursor form exL concurrently with LARP7 and MePCE”.

- First paragraph, page 15: the results from hTR RNA FISH are overstated, as the difference in nuclear versus cytoplasmic hTR between the shRNA control and cells depleted of PRAN, LARP7 or MePCE is not significant. This should be mentioned in the text.

We agree with the reviewer. We changed the statement to read” Although increased fractions of cytoplasmic hTR were observed upon LARP7 (45%) and MePCE (38%) knockdown, none of these differences were statistically significant (Fig. 6g and 6h).”

- Figure 2c: The immunoprecipitations using antibodies against LARP7 or MePCE reveal that other LARPs co-immunoprecipitate strongly with LARP7 and MePCE. This should be discussed in the main text or figure legend.

We added the statement to read “Immunoprecipitation with the anti-LARP3 antibody coprecipitated LARP7 and MePCE (Fig. 2c, lane 7) and vice versa (Fig. 2c, lanes 8 and 9), but did not significantly coprecipitate DKC1.”

- Figure 6h: For the sh-LARP7, the sum of the white and black bars is more than 100%. Also, correct the title of Y axis: "The percentage of hTR localization"

We have corrected the sum of bars and typo.

Reviewer #2 (Remarks to the Author):

In this revised manuscript the authors have largely addressed my concerns and added a significant number of additional experiments. In particular the inclusion of the disease mutant of TR that disrupts the triple helix provides additional insight. There are some remaining concerns below regarding statistical analysis when experimental results are stated to be different, and the manuscript needs to be carefully edited for syntax and clarity.

We thank the reviewer for complementing our work and the thoughtful suggestions on how to improve specific aspects. We have modified the manuscript accordingly and addressed the individual points below.

1. Was the telomerase activity data normalized to the amount of TR present? From the northern blot in figure 1d it appears as if there is a higher level of mature length TR (even though the quantification only indicates a 10% increase). A higher amount of TR could explain the increase in activity. Were the northern blots to analyze TR levels carried out for each biological replicate of the telomerase production to normalize each individual experiment?

We don't do the data normalization to the amount of TR. We did do three biological replicates of TR levels by Northern blotting (Fig. 1d, and Supplementary Figure 2a). Actually, the quantification indicates a less than 10% increase, because we rounded 1.05, 1.06, and 1.09 to 1.1 (Please check Source Data_1 for exact number). Therefore, these 5~9% increase did not significantly affect our consequence.

2. Figure 3d the difference between the amount of mature TR formed with U460C compared to wildtype needs to be tested for significance or stated to be similar to WT. U460C appears to only slightly change the amount of mature TR formed.

We have added statistical analysis as shown in new Figure 3d. The amount of mature TR with U460C mutation significantly increases.

3. Telomerase activity in figure 4e needs to be quantified like in figure 1 including replicates and normalizations, if the authors want to state LARP3 KD increased telomerase activity.

We knocked down LARP3 in 293T cells using three different sh-RNAs targeting different regions of LARP3. As shown in new supplementary Figure 5d, telomerase activity increased when telomerase was purified by immunoprecipitation with DKC1, although the difference was statistically significant.

4. Figure 5C needs statistical analysis.

We have added statistical analysis as shown in new Figure 5C.

5. Telomerase activity in figure 5E needs statistical analysis.

We have added statistical analysis as shown in new Figure 5F.

6. Figure 6 D and E need proper Y axis labels stating what is being measured. Fold change of what?

We have added description.

7. Page 13 line 21 I believe should reference 6e instead of 6d.

We have corrected.

8. Could figure 6f be presented in a way that groups all reads longer than 451 nt to appreciate the differences more easily?

We agree with the reviewer. We now present additional bar graphs showing the fold change in new supplementary Figure 8d for RNA species ending between position 450 and 461.

9. Figure S9 is discussed quite extensively in the text. The authors should consider moving this data to a main figure.

Because data shown in Figure S9 are to support the Figure 6, therefore we prefer to keep this figure in the supplemental part.

10. Figure 6h the LARP7 bar graph has a typo. Bottom should read 45% not 55%.

We have corrected the sum of bars and typo.

11. Abstract needs to be edited for language, it is hard to follow.

Due to a limit on word (150 words or fewer), we have done our best.

12. Page 7 line 21: It is unclear what “these” is referring to. Please restate the subject of this sentence for clarity.

We have changed the statement to read “the 3’ truncated versions of hTR may also contribute to the binding of DHX36”.

13. Page 8 lines 20-23 mentions “components” several times referring to the LARP protein family members. This manuscript mentions many telomerase components, referencing back to prior sentences with such language is very confusing, especially for non-expert readers. Unless it is completely unambiguous the names of the specific proteins should be restated in each sentence.

We agree with the reviewer. We changed “components” to “the name of specific proteins” as statement “LARP3, LARP7, and MePCE disassociated from hTR when the mature hTR forms were produced”

14. Page 10 line 5 “in contrast” could be replaced by “in addition”.

We have corrected.

15. Figure 4 panels are referenced out of order and appear out of order in the figure. Panel C is below Panel D. Please re-order to make it easier to follow.

We have re-ordered it.

16. Page 12 line 16 the sentence starting with “Given” is incomplete, it lacks a main clause.

We have rephrased the sentence.

Reviewer #3 (Remarks to the Author):

Kao and colleagues present a revised manuscript with a number of new experiments and analyses requested by this reviewer and others. Many points are addressed, statistical tests have been added, and some arguments have been strengthened. However there continue to be conclusions not fully supported by data, including some of the new statistical analyses. Overall the manuscript presents an interesting and intricate study on hTR biogenesis, but more careful interpretation is required.

We thank the reviewer for complementing our work and the thoughtful suggestions on how to improve specific aspects. We have modified the manuscript accordingly and addressed the individual points below.

Major points:

1) Regarding the original concern of the conclusion “LARP3 competes with tertiary structure”, the authors now provide a boxH deletion that cannot interact with the UUU end and find increased LARP3 association and hTR degradation rather than maturation, leading to re-statement of the conclusion that LARP3 competition with tertiary structures competes with maturation. A simpler alternative explanation that must be considered in this mutant is that boxH disruption leads to lack of DKC1 binding and increased susceptibility to exonucleases. However, in Fig. 3C, loss of the boxH appears to increase DKC1 pulldown in the mature form, going against the literature and structural data on H/ACA RNPs. This warrants explanation.

The reviewer interprets our results from Figure 3C that “loss of the boxH appears to increase DKC1 pulldown in the mature form, going against the literature and structural data on H/ACA RNPs”. We think that the interpretation should be careful. Fig. 3C only represents the proteins that associate with hTR, but not the species of hTR. After a 120-min incubation of 375-377GGAdel hTR, **the mature form of hTR** substantially decreased as shown in the Figure 3d (lane 20). More dramatic effects could be observed when the reaction was incubated for 60 min. Our data suggest that the 375-377GGAdel mutant reduces accumulation of the mature form of hTR in vitro and support the previous observations in Collins lab that the H box motif of hTR is important for RNA stability and 3' end processing¹. But, previous study in 1999¹ of Collins lab did not show whether the H box mutant affect the H/ACA complex binding.

However, in vitro study published in 2010 of Collins lab has shown that assembly of two sets of H/ACA proteins only need a minimal 5' hairpin stem of H/ACA motif and the H/ACA complex assembly on the 3' hairpin of H/ACA motif does appear to require this large surface of protein-RNA contact². As shown in Fig. 2b in our study, deletion of the 3' stem loop abolished the binding of DKC1 (Fig. 2b, lane 11). Although the 3' stem loop deleted mutant retains the 5' stem loop and the H box, but could not support the DKC1 binding, underscoring the importance of the 3' hairpin of hTR and suggesting an additional role for the box H in mediating hTR maturation.

As the reviewer noticed, our data indicate that loss of the box H appears to increase DKC1 pulldown (Fig. 3C). Our data suggest that the 375-377GGAdel mutant disrupts the formation of triple helix and causes an increase in LARP3 binding. Given that LARP3 preferentially binds to the terminal U stretch of RNA, the binding of LARP3 would prevent both the 3' end processing and the 3'-to-5' degradation of exL. In this regard, it raises a possibility that hTR might be degraded in the 5'-to-3' direction. If this were the case, the increase in DKC1 binding to the 375-377GGAdel mutant would increase. In the future, it will be intriguing to understand the detailed mechanisms of how the H box mutants destabilize hTR in concert with LARP3. Given that we don't have much data to support any mechanisms, we prefer to let the primary data speak for itself.

Reference paper:

1. Mitchell, T.R. et. al., A Box H/ACA Small Nucleolar RNA-Like Domain at the Human Telomerase RNA 3' End. *Molecular and cellular biology*, Jan. 1999, p. 567–576.
2. Egan, E.D., et al., Specificity and Stoichiometry of Subunit Interactions in the Human Telomerase Holoenzyme Assembled In Vivo. *Molecular and cellular biology*, June 2010, p. 2775–2786.

2) New data and statistical analyses do not fully support statements in the rebuttal and text in a number of cases. Conclusions drawn should reflect this. Examples:

a. Fig 6e: northern blot of LARP7 kd is not significant (ns), but text states there are reproducible reductions in the steady-state level of hTR.

We agree with the reviewer. In terms of LARP7 kd, we must carefully interpret the data. Therefore, we deleted the statement.

b. Fig 6f: replicates and statistics not provided to support the statement that LARP7 and MeCPE kd resulted in increased fraction of longer transcripts beyond 451 nt

We now present additional bar graphs showing the fold change in new supplementary Figure 8d for RNA species ending between position 450 and 461. We did not do the replicates. On one hand, we analyzed between 4.7 million and over 6.1 million telomerase RNA reads per sample, which is substantially more than many comparable studies in the literature. On the other hand, the results of PARN kd sample indeed reproduced the same phenotype as published in previous studies.

Reference paper:

1. Tseng, C.K., Wang, H.F., Schroeder, M.R. & Baumann, P. The H/ACA complex disrupts triplex in hTR precursor to permit processing by RRP6 and PARN. *Nat. Commun.* 9, 5430 (2018)
2. Moon, D.H. et al. Poly(A)-specific ribonuclease (PARN) mediates 3'-end maturation of the telomerase RNA component. *Nat. Genet.* 47, 1482–8 (2015).

c. Fig 6g,h: subcellular localization quantification is 'ns' across the board but regarded as different in the text

We agree with the reviewer. To support these results, we did cell fractionation and checked the localization of hTR by Northern blot. As shown in the figure 6i, the fraction of cytoplasmic hTR increased upon PARN, LARP7, and MeCPE knockdown. These data support our image experiments.

d. Argument is made in the rebuttal and revised text for LARP3 level differences explaining inconsistency of results between 293T and K562 ("CML") cells from LARP3 kd, but then it is stated in the rebuttal that levels were found to be no different

The inconsistency of results between 293T and K562 cells from LARP3 kd could be cell line-specific, although the expression levels of LARP3 in these two cell lines are similar when we normalized to the corresponding control (Tubulin). In terms of 293T cells, to our knowledge, no report suggests that the evaluated levels of LARP3 was observed in 293T cells. In contrast, LARP3 expression is correlated with poor clinical prognosis in CML patients and increases during CML progression (chronic phase, the accelerated phase, and blast phase)¹. K562 is characterized by the presence of the BCR-ABL1 fusion gene, which serves as a crucial molecular marker in the diagnosis and treatment of CML in clinical practice². A study published by Calabretta lab gave the first hint about the molecular mechanism leading to elevated levels of LARP3 protein in BCR-ABL1-transformed cells². These findings suggest to us that the **negative effect of LARP3 has been established in K562 cells, but not in 293T cells**. Our data here indicate that overexpression of LARP3 impairs hTR biogenesis, suggesting that high expression levels of LARP3 have negative effect on telomere homeostasis. Indeed, reducing expression levels of LARP3 in K562 cells cause telomere elongation. It is always difficult to compare the same protein between two different types of cell lines. Even these two cell lines express the same level of protein, they may not present the same phenotype.

Reference paper:

1. Trotta, R. et al. BCR/ABL activates mdm2 mRNA translation via the La antigen. *Cancer Cell* 3, 145–60 (2003).
2. Druker, B. J. et al. Effects of a selective inhibitor of the Abl tyrosine kinase on the growth of Bcr-Abl positive cells. *Nat. Med.* 2, 561–566. (1996).
3. Sommer, G., and Heise, T. (2021). Role of the RNA-binding protein La in cancer pathobiology. *RNA Biol.* 18, 218–236. 10.1080/15476286.2020.1792677.

Other points:

1) More description should be provided in the methods or figure legends about the generation and validation of stable shRNA kd and overexpression in cell lines used for telomere length analyses. If these are all done by plasmid transfection and drug selection, for instance, there could be bottle-neck effects of selecting very few cells with integrations that confound bulk telomere length interpretation

We always culture and maintain the knockdown cells in the medium containing $2 \mu\text{g ml}^{-1}$ puromycin for the selection as described in the methods.

2) The sentence structure of new text on page 12 should be corrected

We have corrected.

3) In Supp Fig 1c, I believe the right panel truncation label (3') is incorrect

We have corrected.

4) In the methods, "Preparation of capped hTR...", it is stated that the reaction contains both bioUTP and ^{32}P -UTP. Please confirm.

Yes, we added both bio-UTP and ^{32}p -UTP. The incorporation of small amounts of ^{32}p -UTP in the RNA allows us to perform gel-purification of full length of hTR.

REVIEWERS' COMMENTS

Reviewer #2 (Remarks to the Author):

The authors have addressed all of my concerns and I support publication.

REVIEWER COMMENTS

Reviewer #2 (Remarks to the Author):

The authors have addressed all of my concerns and I support publication.

We thank the reviewer for complementing our work and support for publication.